# ATTRIBUTE-TO-DELETE: MACHINE UNLEARNING VIA DATAMODEL MATCHING

**Kristian Georgiev**[1*], **Roy Rinberg**[2,3*], **Sung Min Park**[4*], **Shivam Garg**[5*]
**Andrew Ilyas**[6], **Aleksander Mądry**[1], **Seth Neel**[2]

[1]MIT EECS    [2]Harvard Business School    [3]Harvard SEAS    [4]Stanford CS
[5]Microsoft Research    [6]Stanford Statistics

## ABSTRACT

Machine unlearning—efficiently removing the effect of a small "forget set" of training data on a pre-trained machine learning model—has recently attracted significant research interest. Despite this interest, however, recent work shows that existing machine unlearning techniques do not hold up to thorough evaluation in non-convex settings. In this work, we introduce a new machine unlearning technique that exhibits strong empirical performance even in such challenging settings. Our starting point is the perspective that the goal of unlearning is to produce a model whose outputs are *statistically indistinguishable* from those of a model re-trained on all but the forget set. This perspective naturally suggests a reduction from the unlearning problem to that of *data attribution*, where the goal is to predict the effect of changing the training set on a model's outputs. Thus motivated, we propose the following meta-algorithm, which we call Datamodel Matching (DMM): given a trained model, we (a) use data attribution to *predict* the output of the model if it were re-trained on all but the forget set points; then (b) *fine-tune* the pre-trained model to match these predicted outputs. In a simple convex setting, we show how this approach provably outperforms a variety of iterative unlearning algorithms. Empirically, we use a combination of existing evaluations and a new metric based on the KL-divergence to show that even in non-convex settings, DMM achieves strong unlearning performance relative to existing algorithms. An added benefit of DMM is that it is a meta-algorithm, in the sense that future advances in data attribution translate directly into better unlearning algorithms, pointing to a clear direction for future progress in unlearning.

## 1 INTRODUCTION

The goal of machine *unlearning* is to remove (or "unlearn") the impact of a specific collection of training examples from a trained machine learning model. Initially spurred by regulations such as the EU's *Right to be Forgotten* (Ginart et al., 2019), machine unlearning has found a variety of recent applications including: removing the effect of toxic, outdated, or poisoned data (Pawelczyk et al., 2024b; Goel et al., 2024); rectifying copyright infringement in generative models (Liu, 2024; Dou et al., 2024); and even LLM safety training (Li et al., 2024; Yao et al., 2024).

This plethora of potential applications has prompted a growing line of research into better *unlearning algorithms*. An unlearning algorithm takes as input a model $\theta$ (trained on a dataset $S$) and a "forget set" $S_F \subset S$, and outputs a model $\theta_{UL}$ that "looks like" it was trained on the so-called "retain set" $S_R := S \setminus S_F$. Of course, one valid unlearning algorithm simply ignores the trained model $\theta$ and trains a new model $\theta_{UL}$ from scratch on the retain set $S_R$. This algorithm clearly succeeds at the task of unlearning, since the generated $\theta_{UL}$ really *is* trained only on the retain set. But as model and dataset sizes continue to increase, or unlearning requests become more frequent, this approach becomes infeasible. The goal of unlearning is thus to *approximate* this naive retraining algorithm while imposing a much lower computational burden.

---

*Equal contribution

For convex models (i.e., models obtained by empirical risk minimization over a loss convex in parameter $\theta$), there are fast unlearning algorithms that also enjoy provable guarantees (Neel et al., 2021; Graves et al., 2021; Izzo et al., 2021b; Mu & Klabjan, 2025; Qiao et al., 2025)

For large neural networks, however—where efficient unlearning is arguably most relevant, given the cost of training from scratch—the situation is considerably murkier. The only methods that obtain provable guarantees tend to significantly degrade accuracy and/or require significant changes to the training pipeline (Bourtoule et al., 2020a; Li et al., 2022). As a result, unlearning algorithms for neural networks typically rely on heuristic approaches that fine-tune an initial model $\theta$ into an "empirically unlearned" model $\widehat{\theta_{UL}}$. These approaches, however, have not yet led to consistently reliable unlearning algorithms, as evidenced by a variety of empirical evaluations and benchmarks (Hayes et al., 2024; Kurmanji et al., 2023; Pawelczyk et al., 2023). In particular, recent evaluations such as U-LiRA (Hayes et al., 2024) demonstrate that the predictions of the empirically unlearned model are often easily distinguishable from the "oracle" predictions by an adversary.

A pervasive challenge for fine-tuning-based approaches is what we refer to as the *missing targets* problem. In order to "unlearn" a forget set point $x \in S_F$, fine-tuning-based methods typically employ some version of *gradient ascent* on $x$, starting from $\theta$, and gradient descent on the retain set $S_R$ in order to maintain performance. If left unrestricted, gradient ascent will continue to make the loss on $x$ arbitrarily high—what we want, however, is to increase the loss only until it reaches its *counterfactual value*, i.e., the loss on $x$ of a model trained on the retain set $S_R$. Ideally, we could terminate the algorithm when the model's loss on $x$ reaches this "target" value, but the problem is that (a) we do not have access to the target; and (b) the optimal "stopping time" might be different for different points $x \in S_F$. The result is the well-documented phenomenon of unlearning algorithms "undershooting" and "overshooting" the loss on different examples (Hayes et al., 2024).

**This work.** In this paper, we present a new unlearning algorithm that sidesteps the issue discussed above, and (empirically) achieves state-of-the-art unlearning performance. Our algorithm resembles prior techniques in that we rely on fine-tuning the trained model $\theta$. We deviate from prior work, however, through two main ideas:

1. **Oracle Matching (OM).** Consider the following thought experiment: what if we could access the *outputs* (but not the parameters) of a model trained on the retain set $S_R$? We show that such "oracle" access directly enables an efficient, fine-tuning-based unlearning algorithm. Rather than minimizing/maximizing loss on the retain/forget sets, this algorithm directly minimizes the difference between model outputs and oracle outputs on a subsample of the train set, thus sidestepping the aforementioned "missing targets" problem. Empirically, we find that the fine-tuned model also *generalizes* beyond the fine-tuning points, and in some way "distills" the target model into parameters $\theta'$.

2. **Oracle simulation.** OM on its own is not an unlearning algorithm—it relies on the very "oracle model" that it aims to replicate. Observe, however, that implementing OM does not require access to the weights of an oracle model, but only to its outputs on a fixed number of inputs. Thus, OM can be implemented efficiently given access to an efficient routine for computing such outputs. Such a routine is precisely the target of *predictive data attribution* methods (Ilyas et al., 2024), where the goal is exactly to predict how a model's outputs would change if its training dataset were modified. This leads to our second idea: instead of fine-tuning on "oracle" outputs, we fine-tune on *simulated* outputs from a predictive data attribution method. We show that despite these methods being imperfect, applying our OM algorithm to *simulated* oracle outputs works nearly as well as using the true oracle outputs.

The resulting algorithm, *datamodel matching* (DMM), not only achieves current state-of-the-art performance (Figure 1), but also introduces a reduction from unlearning to data attribution, allowing us to translate future improvements in the latter field to better algorithms for the former.

The rest of our paper proceeds as follows. In Section 2, we formally introduce the unlearning problem, as well as the field of (predictive) data attribution. In Section 3, we strengthen existing unlearning evaluation by introducing a new metric called *KL Divergence of Margins* (KLoM). KLoM directly adapts a formal definition of unlearning (Neel et al., 2021) to be computationally and statistically tractable to estimate, and addresses some challenges faced by existing unlearning metrics. Then, in Section 4, we combine the two insights above (Oracle Matching and Oracle Simulation) into a simple algorithm called *datamodel matching* (DMM). Finally, in Section 5, we provide some theoretical

justification for our algorithm using a case study of underdetermined ridge (linear) regression. In particular, we show that in this simple setting, the oracle matching (OM) primitive can provably lead to faster convergence. We conclude with a discussion of limitations and directions for future work.

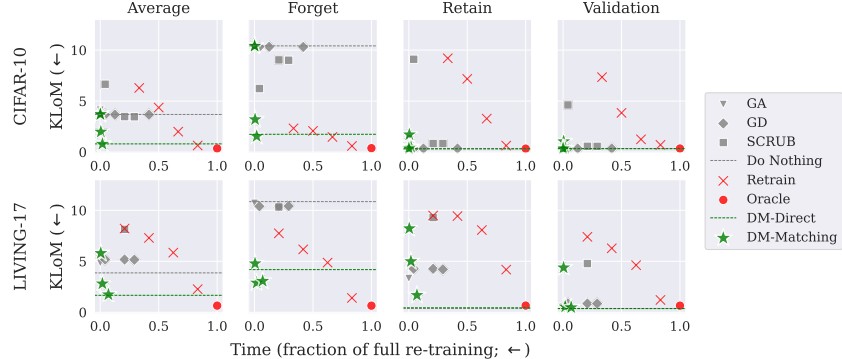

Figure 1: **Leveraging predictive data attribution enables effective unlearning.** We apply different approximate unlearning methods to trained DNNs to unlearn forget sets from CIFAR-10 and ImageNet-Living-17. We measure unlearning quality using KLoM scores (y-axis), which quantifies the distributional distance between unlearned predictions and oracle predictions (0 being perfect). To contextualize each method's efficiency, we also show the amount of compute relative to full re-training (x-axis). We evaluate KLoM values over points in the forget, retain, and validation sets to ensure that unlearning is effective across all datapoints, and report the 95th percentile in each group; we also report their weighted average (1st column). Our new methods based on data attribution (DM-DIRECT and DMM) dominate the pareto frontier of existing unlearning methods, and approach the unlearning quality of oracle models (full re-training) at a much smaller fraction of the cost.

## 2 PRELIMINARIES

In this section, we introduce some preliminary notation, definitions, and results. Throughout the section, we will let $S \in \mathcal{X}^n$ be a fixed dataset drawn from an example space $\mathcal{X}$, and we define a *learning algorithm* $\mathcal{A} : \mathcal{X}^* \to \Theta$ as a (potentially random) function mapping from datasets to machine learning models $\theta$. Finally, for an example $x \in \mathcal{X}$, we use $f_x : \Theta \to \mathbb{R}^k$ to denote a *model evaluation* on the example $x$ (for example, this may be the $k$-dimensional per-class probabilities).

### 2.1 MACHINE UNLEARNING

Consider a machine learning model $\theta \sim \mathcal{A}(S)$ trained on a dataset $S$. Given a "forget set" $S_F \subset S$, and a corresponding "retain set" $S_R = S \setminus S_F$, the goal of an *exact* unlearning algorithm is to compute a sample from $\mathcal{A}(S_R)$ starting from the trained model $\theta$:

**Definition 1** (Exact unlearning (Ginart et al., 2019)). *An unlearning algorithm $\mathcal{U} : \Theta \times 2^{|S|} \to \Theta$ is said to be an exact unlearning algorithm if, for all $S_F \subset S$, $\mathcal{U}(\mathcal{A}(S), S_F) \overset{d}{=} \mathcal{A}(S_R)$, where $\overset{d}{=}$ represents equality in distribution over models.*

Though compelling in theory, exact unlearning tends to be too stringent a criterion when applied to deep learning, often leading to computational infeasibility or degradations in accuracy (Liu, 2024). This motivates a look at *approximate unlearning*, which asks only for the distribution over unlearned models to be $(\epsilon, \delta)$-indistinguishable from re-training:

**Definition 2** (($\varepsilon, \delta$)-unlearning (Neel et al., 2021)). *$\mathcal{U}$ is an $(\epsilon, \delta)$-approximate unlearning algorithm if, for all $\mathcal{O} \subset \Theta, S_F \subset S$ we have that*

$$\Pr\left[\mathcal{U}(\mathcal{A}(S), S_F) \in \mathcal{O}\right] \le e^\epsilon \Pr\left[\mathcal{A}(S_R) \in \mathcal{O}\right] + \delta, \tag{1}$$
$$\Pr\left[\mathcal{A}(S_R) \in \mathcal{O}\right] \le e^\epsilon \Pr\left[\mathcal{U}(\mathcal{A}(S), S_F) \in \mathcal{O}\right] + \delta$$

This definition (intentionally) resembles differential privacy (Dwork & Roth, 2014), and asks for the distribution of unlearned models to be statistically close to the distribution of re-trained "oracle"

models. In particular, this condition guarantees than an adversary who observes the model returned by the unlearning algorithm $\mathcal{U}$ cannot draw any inferences with accuracy that is much higher than if the model was fully re-trained.

While unlearning algorithms achieving Definition 2 exist for convex models (Neel et al., 2021; Izzo et al., 2021b; Guo et al., 2019), and for non-convex models when the training process is altered or under stylized optimization conditions (Bourtoule et al., 2021; Chien et al., 2024; Gupta et al., 2021), the bulk of ongoing work in unlearning evaluates Definition 2 *empirically*, rather than as a provable property. We return to the problem of evaluating unlearning algorithms more carefully in Section 3.

## 2.2 PREDICTIVE DATA ATTRIBUTION (DATAMODELING)

Our work also draws on a separate line of work in machine learning called *data attribution* (Koh & Liang, 2017; Hammoudeh & Lowd, 2024; Ilyas et al., 2024). Broadly, data attribution is an area concerned with connecting training data samples to the predictions of the corresponding ML models. Of particular relevance to our work is a particular type of data attribution called *predictive data attribution* (also known as datamodeling (Ilyas et al., 2022; Park et al., 2023)).

In predictive data attribution, the goal is to produce an estimator (or *datamodel*) that takes as input a training set, and as output accurately predicts the behavior of a machine learning model trained on that training set. Using our existing notation: for an example $x \in \mathcal{X}$, a datamodel for $x$ is a function $\hat{f} : 2^S \to \mathbb{R}^k$ such that, for any $S' \subset S$,

$$\hat{f}_x(S') \approx f_x(\mathcal{A}(S')). \tag{2}$$

In other words, $\hat{f}_x(S')$ directly predicts the result of applying the training algorithm $\mathcal{A}$ to the dataset $S'$, and evaluating the function $f$ on the resulting model. Despite the complexity of modern training algorithms $\mathcal{A}$ (e.g., training deep neural networks with stochastic gradient descent), Ilyas et al. (2022) empirically show that *linear* datamodels often suffice to accurately predict model behavior. In other words, for an example $x$, one can compute a vector $\beta \in \mathbb{R}^{|S|}$ such that, for subsets $S' \subset S$,

$$\hat{f}_x(S') := \sum_{z_i \in S'} \beta_i \approx f_x(\mathcal{A}(S')).$$

To compute these coefficients, Ilyas et al. (2022) sample a variety of subsets $S_1, \ldots, S_k$ at random from $S$, and then solve the (regularized) regression problem

$$\beta = \min_{w \in \mathbb{R}^n} \frac{1}{m} \sum_{i=1}^{m} (w^\top \mathbf{1}_{S_i} - f_x(\mathcal{A}(S_i)))^2 + \lambda\|w\|_1. \tag{3}$$

They show that despite the datamodel being constructed using random subsets $S_i \subset S$, the function $\hat{f}$ remains remarkably accurate on non-random datasets (see Ilyas et al. (2022) for a full evaluation). Linear datamodels are particularly appealing for two reasons. First, the coefficients $\beta_i$ have an intuitive interpretation as the influence of the $i$-th training example on a model's prediction on $x$ (Koh & Liang, 2017; Feldman, 2021). Second, they establish a connection to a class of statistical techniques relating to influence functions, which has unlocked a suite of tools for estimating the coefficients more effectively (Park et al., 2023; Grosse et al., 2023).

## 3 EMPIRICALLY EVALUATING UNLEARNING

In Section 2, we introduced the unlearning problem, culminating in a formal definition of the problem (Definition 2). As stated, evaluating whether Definition 2 holds for a given unlearning algorithm is a difficult problem for several reasons. First, given the overparameterized nature of large-scale models, fully satisfying Definition 2 is likely impossible, and verifying it involves comparing distributions in a space with millions or billions of dimensions. Secondly, the definition generally needs to hold over arbitrary forget sets $S_F$, or at least across a range of forget sets $S_F$ likely to occur in practice.

**The current evaluation paradigm.** To deal with these problems, one typically evaluates unlearning by focusing on model *outputs* $f_x$[1] rather than model parameters, and testing for the *implications* of

---

[1] A common choice of model output used in prior work, which we will also use, is the *margin* of the classifier.

Definition 2 rather than for the definition directly. In particular, the strongest existing unlearning evaluation for supervised learning, called U-LiRA (Hayes et al., 2024), takes inspiration from *membership inference attacks* (MIAs) (Carlini et al., 2022) and evaluates the ability of an adversary to distinguish between the distribution of outputs of an unlearned model on (a) validation examples and (b) unlearned examples.

**A more direct evaluation.** Recall from Section 2 that traditionally the target of unlearning algorithms has been $(\varepsilon, \delta)$-approximate unlearning (Definition 2). Note that in essence, Definition 2 simply asks for the distribution induced by the unlearning algorithm be "close" to a distribution of models that have never been trained on $S_F$. In particular, we can view it as a special case of the condition

$$\Delta_\delta(\mathcal{U}(\mathcal{A}(S), S_F), \text{safe}(S_F)) \leq \epsilon, \tag{4}$$

where $\Delta_\delta$ is a statistical divergence measure parameterized by $\delta > 0$, and $\text{safe}(S_F)$ is a distribution of "safe" models (i.e., models that have not been trained on $S_F$). We can recover Definition 2 exactly by letting $\text{safe}(S_F)$ be exactly $\mathcal{A}(S \setminus S_F)$, i.e., the distribution of models trained on all but the forget set and setting $\Delta_\delta$ appropriately.

We make two observations about (4). First, the choice of $\text{safe}(S_F)$ is somewhat arbitrary, and in particular *any* distribution that does not depend on $S_F$, and produces a useful model would suffice. This includes, for example, distributions $\mathcal{A}'(S \setminus S_F)$ for algorithms $\mathcal{A}' \neq \mathcal{A}$, or distributions of *ensembles* of models $\mathcal{A}(S \setminus S_F)$. Second, while the $\Delta_\delta$ used in Definition 2 has an appealing privacy interpretation, it is sensible (especially given our focus on empirical evaluation) to consider other divergences that are easier to estimate. These two observations inspire a metric that we call KLoM for empirical unlearning evaluation. KLoM corresponds to Definition 2 where we (a) use $\Delta = \text{KL}$ divergence, (b) allow for an arbitrary "reference distribution" $\text{safe}(S_F)$, and (c) as in U-LiRA, study distributions of model *outputs* $f_x$ rather than parameters.

**Definition 3** (KL divergence of margins (KLoM)). *For an unlearning algorithm $\mathcal{U}$, reference distribution $\text{safe}(S_F)$, and input $x$, the KL divergence of margins (KLoM) is given by*

$$\text{KLoM}(\mathcal{U}, \text{safe}(S_F), x) := D_{KL}(\text{safe}(S_F), \ f_x(\mathcal{U}(\mathcal{A}(S), S_F))) \,.$$

Despite the arbitrariness of $\text{safe}(S_F)$, unless otherwise noted we will mirror Definition 2 and take $\text{safe}(S_F) := \mathcal{A}(S \setminus S_F)$. Throughout the rest of this work, we primarily evaluate unlearning algorithms via computing KLoM for different inputs $x$ from the forget set, retain set, and validation set. We also evaluate our algorithms with U-LiRA, and defer these results to the Appendix.

Compared to U-LiRA, KLoM is simpler to implement, has a natural correspondence with our original Definition 2, and importantly, does not suffer from *catastrophic unlearning*: observe that an unlearning algorithm $\mathcal{U}$ that transforms its input into a random classifier will pass an U-LiRA evaluation, as the random classifier will treat unlearned points and validation points identically. In contrast, by forcing us to explicitly specify $\text{safe}(S_F)$, KLoM explicitly compares unlearned models to a baseline whose performance we know *a priori*. Crucially, both KLoM and U-LiRA evaluate unlearning algorithms using *point-specific* distributional estimates, which as observed in Hayes et al. (2024) makes these evaluations far more stringent than prior approaches.

## 4 DMM: UNLEARNING BY SIMULATED ORACLE MATCHING

Having defined an evaluation apparatus, we now introduce our algorithm for machine unlearning. We first motivate the algorithm by observing a common challenge in existing methods. We then, in Section 4.2, propose an effective hypothetical algorithm for unlearning, under the unrealistic assumption that we have access to outputs of the "oracle" model. In Section 4.3, we show how to accurately simulate such oracle outputs using data attribution methods. Finally, in Section 4.4, we combine these insights and present our final algorithm, datamodel matching (DMM), and demonstrate its effectiveness and efficiency.

### 4.1 MOTIVATION: THE MISSING TARGETS PROBLEM

Recall that the goal of unlearning is to approximate an *oracle* model, i.e., a model that was never trained on a given "forget set" of data. In strongly convex settings, this oracle model is unique, since

it corresponds to the minimizer of a strongly convex loss function over the complement of the forget set (called the *retain set*). Thus, running gradient descent (GD) on the retain set loss yields a provable (and in some cases, efficient (Neel et al., 2021)) unlearning algorithm.

In the context of deep neural networks, however, GD alone is insufficient. In these settings, the loss function and training data alone do not fully specify the final model. In particular, once we have already minimized loss on the forget set, applying GD on the retain set does not significantly alter forget set predictions, preventing us from recovering the oracle model. Many unlearning methods for deep neural networks (Triantafillou et al., 2023; Kurmanji et al., 2023) thus actively *increase* loss on the forget set (e.g., via gradient ascent) while *maintaining* performance on the retain set.

This general approach comes with a significant set of drawbacks, which we collectively refer to as the *missing targets* problem. First, the assumption that forget set points will increase in loss after unlearning and retain set points will not is not necessarily correct. For example, if there are duplicated points across the forget and retain sets, then loss on points in the retain set might increase, while loss on points in the forget set might not change. Second, even for a forget set point whose loss *does* increase under the oracle model, our goal is not to increase loss arbitrarily, but instead only until it reaches its "target value" under the model. Since we lack access to these values, it is challenging to know when a given forget set point has been "unlearned." Prior work tries to address this by devising heuristic regularization schemes, e.g., via early stopping, but nevertheless often overshoot or undershoot the target loss for a given data point. Figure 2 illustrates this phenomenon for a popular unlearning algorithm called SCRUB: over iterations of the algorithm, different points are unlearned (and then subsequently "overshot") at different points in time (Hayes et al., 2024).

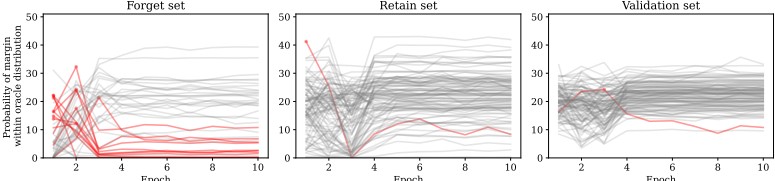

Figure 2: **The missing targets problem.** We apply the SCRUB (Kurmanji et al., 2023) algorithm to unlearn a forget set of CIFAR-10, and measure how well different (random) points are unlearned over time. To quantify how well a given point $x$ is unlearned, we fit a Gaussian distribution to the outputs of oracle models on $x$, and compute the likelihood of the average outputs from unlearned models under this distribution. For many examples in the forget set (shown in red), unlearning quality is hurt by training for too long as we lack access to oracle targets.

### 4.2 THE ORACLE MATCHING ALGORITHM

In essence, the underlying challenge is that we do not know the oracle model's behavior a priori. But what if we did have access to its predictions? In particular,

*Given access to sample outputs from the oracle model (re-trained without the forget set), can we efficiently fine-tune an existing model (trained on the full dataset) to match the outputs out of sample?*

While assuming access to oracle outputs is unreasonable—since our goal is to produce an oracle model in the first place—later in Section 4.4, we will replace oracle access with an efficient proxy using data attribution. For now, we simply assume we have direct access to oracle predictions, and focus on understanding whether gradient-based optimization can match predictions of the oracle. Even in this idealized setting, it is not clear how fast (if at all) gradient descent can converge to an oracle model. For example, whether we can do this efficiently with a small sample is unclear; it is possible that fine-tuning the trained model can match the oracle predictions on the sampled points, but fail to generalize when evaluated on held-out points.

Formally, we assume access to predictions of an oracle model $f^{\text{oracle}}(x) := f_x(\mathcal{A}(S_R))$, where again $S_R$ is the retain set and $f_x$ is the evaluation of the model on input $x$ (e.g., in classification settings one can take $f$ to be the logits of the neural network). The *Oracle Matching* (OM) algorithm runs gradient descentto minimize the MSE between the output logits from the model $f_x(\theta)$ and oracle predictions $f^{\text{oracle}}(x)$ on samples $x$ from the forget and retain sets; see the pseudocode in Algorithm C.1.

**Evaluating oracle matching.** We evaluate OM on various forget sets on two image classification tasks: ResNet-9 models trained on CIFAR-10 and ResNet-18 models trained on an ImageNet subset Living-17 (Santurkar et al., 2020). We compare OM to the following unlearning baselines:[2] gradient ascent (GA) on forget set, gradient descent (GD) on retain set, SCRUB (Kurmanji et al., 2023), a no-op "Do Nothing" baseline, and partial or full re-training. We evaluate all methods using KLoM (Section 3) over distribution of 100 unlearned (method-specific) and 100 re-trained models (see Appendix F for more details).

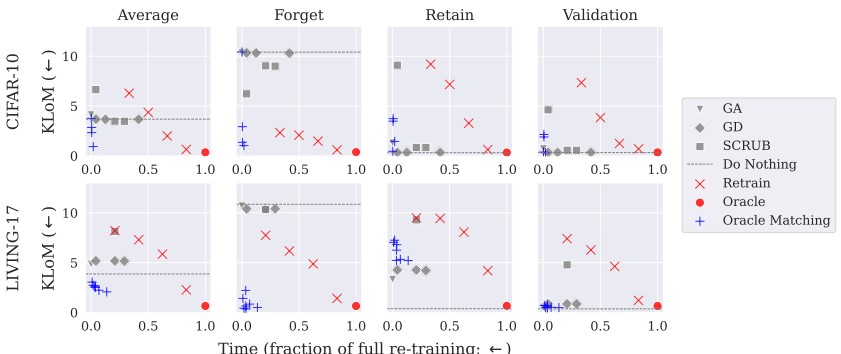

Figure 3: **Oracle matching can efficiently approximate re-training.** The KLoM metric (y-axis) measures the distributional difference between unlearned predictions and oracle predictions (0 being perfect). We also show the amount of compute relative to full re-training (x-axis). We evaluate KLoM values over points in the forget, retain, and validation sets and report the 95th percentile in each group; we also report the average across groups (1st column).

The results (Figure 3) demonstrate that OM is able to efficiently match the predictions of the oracle. Models unlearned with OM closely match the oracle distribution—as measured by KLoM scores—across all splits of the dataset (forget, retain, and validation sets), significantly outperforming all of the prior gradient-based approaches. Importantly, OM achieves effective unlearning while using *less than 5%* of the compute of full-retraining. In contrast, matching the performance of OM on the forget set by retraining requires spending more than $60\%$ of full retraining time. The success of OM implies that for any given trained model $\theta$ and the forget sets we studied: i) there exists another model $\theta'$ close in parameter space that yields similar predictions as an oracle retrained without the forget set; and ii) $\theta$ can be fine-tuned to quickly converge to $\theta'$ with a sufficient sample of oracle outputs. We find that using a sufficiently high ratio of forget points in the fine-tuning set and a sufficient fraction of retain points (but still much smaller than the full train set) is able to provide enough guidance (see Appendix G for exact details).

### 4.3 An efficient proxy for oracles: datamodels

We saw that OM is highly effective at approximating oracle outputs, but OM is not a practical algorithm as it assumes access to oracle outputs. To now turn this into a practical algorithm, we leverage methods for predictive data attribution (introduced in Section 2) to *simulate* oracle outputs.

Recall that a *datamodel* $\hat{f}_x$ predicts the counterfactual output of the model on input $x$ when trained on an arbitrary subset $S \setminus S_F$: $\hat{f}_x(S \setminus S_F) \approx f_x(\mathcal{A}(S \setminus S_F))$. In the case of linear datamodels, we can parameterize the datamodel with a vector $\beta(x)$ so that $\hat{f}_x(S \setminus S_F) := \sum_{i \in S \setminus S_F} \beta_i(x)$. Leveraging linearity, we can re-write this as $\sum_{i \in S} \beta_i(x) - \sum_{i \in S_F} \beta_i(x)$, and we also replace the first term with the starting model output $f_x(\theta_0)$. Our general algorithm, DM-DIRECT (Appendix C.2), simulates the oracle outputs as $h(x) := f_x(\theta_0) - \sum_{i \in S_F} \beta_i(x)$.

**Estimating datamodels.** To estimate datamodels, we follow the approach in (Ilyas et al., 2022): we train models random subsamples of the full training set and use sparse linear regression to fit datamodel vectors $\beta(x)$ (later in Appendix G.2 we explore alternative estimators).

---

[2]See Appendix D.4 for detailed descriptions of each.

**Evaluating DM-DIRECT.** In Figure 4, we compare model outputs on random forget and retain examples; the histograms show that the unlearned outputs from DM-DIRECT closely approximate the true oracle outputs. KLoM evaluations (Figure 1) show that DM-DIRECT (green line) in produces outputs close in distribution to that from oracle re-training for almost all points in the data distribution.

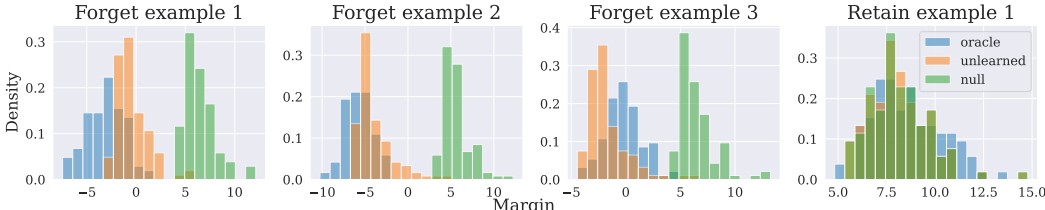

Figure 4: **Datamodels predict oracle outputs.** For random samples from the forget and retain sets, we compare the distribution (across multiple runs) of margins when evaluted on that example across three settings: i) null (model on full dataset); ii) oracle (model re-trained without forget set); and iii) unlearned (using DM-DIRECT). In every case, the predicted outputs (orange) closely match the ground-truth (oracle), demonstrating the effectiveness of datamodels as a proxy for oracle outputs.

## 4.4 ORACLE MATCHING WITH DATAMODELS

Now that we have an efficienty proxy for oracle outputs via datamodels, we revisit the OM algorithm from earlier. Our final algorithm, datamodel matching (DMM), first uses datamodels to generate approximations of oracle predictions on a subset of retain points and the forget points, and then runs OM on the datamodel predictions (see Appendix C.3 for pseudocode).

In Figure 1, we contextualize the performance of DMM against baselines and DM-DIRECT from earlier. DMM achieves levels of unlearning similar to that of fully retraining the model (as measured by KLoM scores), while using significantly less compute.[3] Using datamodels allow us to recover the performance of OM, and outperforms all prior gradient-based approaches. Importantly, DMM also matches the test accuracy of the oracle model and maintains accuracy on the points in the retain set (Appendix H.4), a common failure mode in prior methods. In particular, DMM is significantly more effective than partially re-training given the same computational budget. We do not include the cost of computing datamodels as this is a *one-time* cost and hence amortized over many unlearning requests. [4] This is possible because once a predictive *datamodel* has been constructed (either via re-sampling as done here or influence function-like approximations, which we explore in Appendix G.2), the datamodel generalizes well to new forget sets in practice.

To better understand DMM, in Appendix G, we ablate different components of oracle matching and datamodel estimation. We show that OM addresses the motivating problem of missing targets, leading to stability in optimization. We also show that leveraging more efficient estimators for datamodels such as TRAK (Park et al., 2023) still yields effective unlearning via DMM.

## 5 ORACLE MATCHING FOR LINEAR MODELS

We have seen that empirically OM outperforms standard gradient-based unlearning methods, and we have highlighted the missing targets problem as one possible explanation. Are there other factors that contribute to the success of OM relative to prior methods gradient-based methods, and can we better understand what settings we expect OM to perform well? This motivates studying a setting where the missing targets problem is neutralized: when the objective is strongly convex. In this setting, the unlearned model is the unique empirical risk minimizer on $X_R$, and GD initialized at the current model is a provably effective unlearning algorithm (Neel et al., 2020). Even in the setting when GD on its own can converge, does providing "guidance" from an oracle help?

We answer this affirmitively: First, in Subsection 5.1, empirically identify two factors that influence whether OM outperforms GD: the degree of regularization and stochasticity in optimization. Next, in

---

[3]We only count the finetuning cost; the cost of computing datamodels is amortized across unlearning updates.

[4]The practice of not including pre-computation costs is standard in literature, e.g., Izzo et al. (2021a).

Subsection 5.2, we theoretically characterize the exact convergence rates of full batch OM and GD to the unlearned model in terms of the degree of regularization and the relative eigenmass on the forget and retain sets. Unlike in the full-batch setting where both algorithms converge at a linear rate, in the stochastic setting we show that OM converges exponentially faster than SGD, which sheds light on the superior performance of stochastic OM in our empirical results.

**Setting.** We consider the following ridge regression algorithm, given by

$$\mathcal{A}(S) := \arg\min_{\theta} \sum_{(\boldsymbol{x_i}, y_i) \in S} \left(\theta^\top \boldsymbol{x_i} - y_i\right)^2 + \lambda \|\theta\|_2^2, \tag{5}$$

where $\boldsymbol{x_i} \in \mathbb{R}^d$ are the training inputs, $y_i \in \mathbb{R}$ are the corresponding labels, and the setting is overparameterized, so $d > |S|$. Given a model $\theta_{\text{full}} = \mathcal{A}(S)$ trained on a full dataset $S$, our goal is to unlearn the forget set $S_F \subset S$ by obtaining a model that minimizes the objective on the retain set $S_R = S \setminus S_F$. For convenience, we use $X$, $X_R$, and $X_F$ to denote the covariate matrices for the full dataset $S$, the retain set $S_R$, and the forget set $S_F$ respectively. We choose the under-determined ridge regression for three reasons: (a) The objective (5) is strongly convex, and so GD on $X_R$ is guaranteed to compute the (unique) unlearned model if ran for sufficiently many iterations; (b) the least-squares objective is amenable to theoretical analysis; and (c) the over-parameterized setting is most relevant to modern deep learning models where $d \gg n$.

**Unlearning algorithms.** Let $\theta_* = \mathcal{A}(S_R)$ be the minimizer of the ridge regression objective on the retain set (i.e., the unlearning target). Starting from $\theta_{\text{full}} = \mathcal{A}(S)$, we evaluate several iterative first-order unlearning algorithms in terms of their ability to recover $\theta_*$: (i) GD minimizes the ridge regression objective on the retain set $S_R$ using gradient descent with constant step size, starting from $\theta_{\text{full}}$, (ii) GDA incorporates forget set points in the gradient descent updates, combining gradient descent on the retain set with gradient ascent on the forget set; (iii) OM assumes query access to an unlearned model $\theta^*$, and uses gradient descent (with constant step size) to minimize squared error with respect to "oracle" predictions $\boldsymbol{x_i}^\top \theta^*$ on the full dataset, aiming to minimize $\|X\theta - X\theta_*\|^2$; OMRS performs gradient descent on the squared error from oracle predictions but only on the retain set, thereby minimizing the objective $\|X_R\theta^* - X_R\theta\|^2$. We analyze both the full-batch and stochastic versions of these methods. See Appendix E.3 for details on the setup of the algorithms we evaluate.

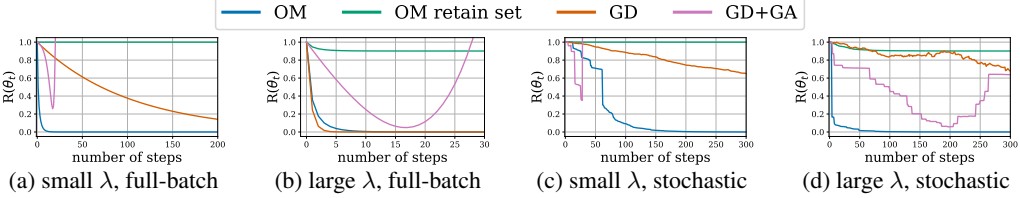

(a) small $\lambda$, full-batch    (b) large $\lambda$, full-batch    (c) small $\lambda$, stochastic    (d) large $\lambda$, stochastic

Figure 5: Comparing unlearning methods for a linear model with $d = 400, n = 100, |S_F| = 5$. The y-axis shows the relative squared distance to the optimal unlearned model $R(\theta_t) = \frac{\|\theta_* - \theta_t\|}{\|\theta_* - \theta_{\text{full}}\|}$, where $\theta_t$ is the iterate at time $t$, $\theta_{\text{full}}$ is the model trained on all data, and $\theta_*$ is the optimal unlearned model.

## 5.1 LINEAR MODEL EXPERIMENTS

Figure 5a depicts the performance of the four unlearning algorithms we consider with low regularization ($\lambda = 4$) in the full-batch setting. We observe that OM converges to the unlearned solution much faster GD, while GDA and OMRS both fail to converge even as $t \to \infty$. Given that OMRS makes negligible progress, we can conclude that the success of OM is due to inclusion of the forget points. Investigating further, we observe that during unlearning, the model parameters change the most in directions orthogonal to the retain set: despite the fact that $24\%$ of the mass of the forget set points lies in the span of the retain set, this span actually captures less than $0.01\%$ of the mass of the ground-truth update $\theta_* - \theta_{\text{full}}$. GD is only able to make progress orthogonally to the retain set due to the the $\ell_2$ regularization term, and when $\lambda$ is low, this rate of progress is slow. On the other hand, OM makes rapid progress in these directions due to the inclusion of forget set points. In Figure 5b, we consider the same setting as above but set $\lambda$ to a much larger value of 400. Here, GD converges slightly faster than OM due to the stronger $\ell_2$ regularization, which aids GD in converging along directions

orthogonal to the retain set. Thus, OM converges faster than GD when $\lambda$ is moderate but can be slower with large $\lambda$. In both settings OMRS and GDA do not successfully converge, but GDA—guided by the heuristic use of the forget set points—initially makes significant progress towards $\theta_*$ before eventually diverging. In Figures 5c and 5d, we replicate the experiments above with stochastic variants of the unlearning algorithms. As in the non-stochastic case, we see the OM on the retain set fails to make any progress, and that GDA makes quick progress but then diverges. However, unlike in the full-batch setting, SOM outperforms SGD in both the large and small $\lambda$ settings (see Appendix E.3 for a discussion). This surprising finding is characterized in Theorem 2 below, where we show that SOM converges exponentially faster than SGD.

## 5.2 CONVERGENCE THEORY

We now turn to studying the algorithms theoretically, starting with the full-batch case, corresponding to Figures 5a and 5b above; then the stochastic/minibatched case (Figures 5c and 5d). In all cases, we will focus on the two convergent algorithms above: oracle matching and ridge gradient descent.

**Full-batch case.** In Theorem 1, we provide a theoretical analysis of the convergence rates of GD and OM. The key takeaway here is that the convergence rate for both algorithms depends on both (a) the relative eigenmass of the forget and retain sets; and (b) the strength of the ridge regularization.

**Theorem 1** (Proof in Appendix E.1). *Let $S$ and $S_R$ be the full training set and the retain set respectively, with input matrices $X$ and $X_R$ and corresponding labels $y$ and $y_R$. Additionally, let $\theta_{full}$ and $\theta_*$ denote the optima of the ridge objective (5) for the full data $S$ and retain set $S_R$ respectively. After $t$ iterations of unlearning starting from $\theta_{full}$, the iterate $\theta_t$ satisfies*

$$\theta_t - \theta_* = \begin{cases} (I - 2\eta\lambda)^t \left( I - \frac{2\eta}{1-2\eta\lambda} X_R^\top X_R \right)^t (\theta_{full} - \theta_*) & \text{for ridge gradient descent (GD).} \\ \left( I - 2\eta X^\top X \right)^t (\theta_{full} - \theta_*) & \text{for oracle matching (OM).} \end{cases}$$

Theorem 1 shows that both (full-batch) OM and GD exhibit linear convergence, albeit at different rates. Indeed, in directions orthogonal to the retain set, the middle term in the GD convergence rate disappears, and so the rate depends only on $\eta\lambda$. Thus, as long as the learning rate $\eta$ is set high enough (i.e., not to cancel out the $\lambda$) higher regularization will cause GD to converge faster.

**Stochastic case.** In our experimental analysis, we saw that unlike the full-batch case, the stochastic version of oracle matching was *consistently* more effective than that of gradient descent. In Theorem 2 we show that, at least in the setting of under-determined ridge regression we consider here, this observation is strongly supported by theory. In particular, we show that while OM converges at a linear rate (i.e., exponentially fast in $t$), we can show a $\Omega(\frac{1}{t^2})$ lower bound on the convergence of SGD, giving a strong separation between the two methods.

**Theorem 2** (Proof in Appendix E.2). *Consider the setting of Theorem 1, where $\theta_t$ is the iterate after $t$ steps of unlearning initialized at $\theta_{full}$. Further let $\gamma_{min}, \gamma_{max}$ denote the minimum and maximum eigenvalues of $X^\top X$, and let $A$ and $B$ be lower and upper bounds on the norm of the covariates (i.e., $B \geq \|x_i\| \geq A$ for all $i$). Then, as long as the learning rate $\eta \in (0, \frac{2}{5(\gamma_{max}+\lambda)})$ and $rank(X) > 1$,*

$$\frac{\mathbb{E}\left[\|\theta_t - \theta^*\|^2\right]}{\|\theta_{full} - \theta_*\|^2} \in \begin{cases} O\left((1 - \gamma_{min}\eta)^t\right) & \text{for oracle matching (OM).} \\ \Omega\left(\frac{1}{t^2}\right) & \text{for ridge gradient descent (GD).} \end{cases} \tag{6}$$

The intuition is as follows. In each update step, stochastic OM updates the model parameters only in the span of the random subset of points used for that update. In contrast, stochastic GD decays the model parameters in other directions due to the regularization term. (Recall that this is what led to GD's improved convergence in the high-regularization setting.) While this shrinkage is beneficial in the subspace orthogonal to the retain set, stochastic GD also decays the parameters along the directions spanned by retain set points that are not in the current batch. So, while increasing $\lambda$ improves full-batch GD's convergence speed, this advantage does not hold for stochastic GD.

## 6 CONCLUSION

In this work, we have shown that reducing unlearning to predictive data attribution yields a general and effective framework for unlearning.

ACKNOWLEDGEMENTS

We thank Jamie Hayes and Ilia Shumailov for discussions on machine unlearning and ULIRA. We thank Salil Vadhan for some useful discussions throughout the paper. RR was supported in part by NSF grant BCS-2218803. Additionally, we acknowledge Harvard SEAS and MIT CSAIL for providing computational resources.

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

# Appendices

## A LIMITATIONS

In this work, we presented a general framework for reducing unlearning to the related problem of predictive data attribution. Our fine-grained evaluations using KLoM—which directly measures the quality of unlearning in terms of the difference in the distributions of the unlearned model's outputs from oracle counterparts—demonstrate that DMM significantly outperforms prior gradient-based unlearning methods and approaches the performance of oracle re-training. To conclude, we discuss some limitations and promising directions for future work:

**Extending techniques and evaluations beyond classification.** While our methods perform well in classification settings with few classes, extending them to work in settings with more classes (e.g., full ImageNet or language-modeling) would make them more practical. Extending our techniques directly (i.e., attributing and estimating all class logits) would incur a heavy computational cost, so additional techniques will be necessary to make the algorithms more scalable.

**Improving and understanding oracle matching.** As we saw in **??**, OM and DMM can sometimes cause a mismatch on the retain set due to "reversing the overfitting." [Roy: todo- provide a reference to where this is defined. ] Better understanding the dynamics of the OM algorithm and leveraging other insights (e.g., from the model distillation literature) would be valuable for making the matching part of our framework more stable. One potential direction is to understand when and how better sampling strategies (instead of random subsampling of the retain set) can improve general matching algorithms. [Sam: + more clever sampling strategies]

**Reducing computational costs.** Even the most efficient data attribution methods require a non-trivial computational cost (at least on the same order as training the original model). Can we design other cheaper alternatives for data attribution that we can still leverage for—and are possibly tailored to—practical unlearning scenarios without the full computational cost?

**Applying to more practical scenarios.** In our analysis, we only considered single unlearning requests (i.e., removing one forget set). A natural way of extending them to multiple unlearning requests is to apply DMM sequentially. However, it is plausible that after too many unlearning updates with DMM, the model diverges far enough so that we need to "recalibrate" the pre-computed datamodels. Analyzing how well existing unlearning algorithms and DMM compose under multiple unlearning requests and other practical scenarios can be valuable for understanding the practicality and failure modes of these methods.

## B RELATED WORK

We provide a high level overview of prior works most relevant to our setting. For a more extensive survey of unlearning, see Liu (2024).

**Machine unlearning: goals and evaluations.** While other lines of work also study unlearning "concepts" or "knowledge" (Ravfogel et al., 2022; Eldan & Russinovich, 2023; Kumari et al., 2023; Zhong et al., 2023), we focus on the data-driven notion of unlearning (Cao & Yang, 2015; Ginart et al., 2019; Wu et al., 2020; Neel et al., 2021; Bourtoule et al., 2020b). Our focus is on approximate unlearning methods. However, other works (e.g., Bourtoule et al. (2020b)) aim for *exact* unlearning (e.g., by careful data partitioning and ensembling or leveraging differential privacy). While these approaches come with provable guarantees, they often come at the cost of accuracy (Bourtoule et al., 2020b), so most unlearning algorithms for deep learning are approximate. This approximate nature necessitates empirical evaluations in lieu of a provable guarantee. One line of work adapts membership inference attacks (MIAs) to evaluate machine unlearning (Golatkar et al., 2020; Goel et al., 2022; Hayes et al., 2024). Complementary to that, Pawelczyk et al. (2024a) evaluate machine unlearning methods' ability to remove backdoor attacks from the training set. More broadly, evaluation in these setting can be nuanced: Thudi et al. (2022) argue that due to the stochastic nature of deep learning optimization, approximate evaluation of machine unlearning is only well-defined on an algorithmic level, and not on an individual model instance level.

**Prior unlearning approaches in deep learning.** Due to challenges of developing rigorous unlearning methods in non-convex settings, typical approaches involve some form of gradient-based

optimization. Strategies include: partial fine-tuning (Goel et al., 2022), combinations of gradient descent and ascent (Kurmanji et al., 2023), and sparsity-regularized fine-tuning (Jia et al., 2024) among others. Our approach also employs fine-tuning, but differs primarily in that we address the common problem of "missing targets." Other approaches employ parameter updates based on a local quadratic approximation (Golatkar et al., 2020; **?**) or influence function approximations (**?**); these can be interpreted as also leveraging different forms of data attribution. However, our approach is unique in its use of predictive data attribution only as *guidance* and still employing fine-tuning, which is a more flexible and robust strategy than direct parameter updates.

The primary method we compare against is SCRUB (and SCRUB+R) as it achieves the current state-of-the-art on strong unlearning evaluations Hayes et al. (2024). At a high level, SCRUB finetunes the original model to: i) maximize the KL divergence between the probabilities of the original model and the new model on forget points; ii) minimize the KL divergence on retain points; and iii) also minimize test loss. Despite their alternative design choices from other fine-tuning approaches (e.g., use of KL divergence), our analyses suggest that it suffers from similar underlying challenges.

**Data attribution.** Key to our framework is a reduction to the problem of predictive data attribution (Ilyas et al., 2022; Park et al., 2023). More broadly, the problem of attributing model predictions back to training data has been extensively studied in recent machine learning literature (Koh & Liang, 2017; Park et al., 2023; Engstrom et al., 2024; Grosse et al., 2023; Choe et al., 2024; Bae et al., 2024), with some of the ideas originating from statistics (Hampel, 1974; Jaeckel, 1972; Pregibon, 1981). For an extensive survey on the topic, refer to Hammoudeh & Lowd (2024); Ilyas et al. (2024).

**Model distillation.** Our approach of fine-tuning on (simulated) oracle predictions has some similarity to a different line of work on *knowledge distillation* (**?**), e.g., distilling an existing "teacher" model (possibly an ensemble) to a "student" model (often smaller). We can cast oracle matching as distilling an oracle model into the current model. The main difference, however, is that in our setting, the model we fine-tune is already trained on the full dataset, and our goal is only to apply a small update to this model.

## C PSEUDOCODE

### C.1 ORACLE MATCHING

---

**Algorithm C.1** Oracle Matching (OM)

---

1: **Input:** Trained model $\theta_0$; oracle predictions $f^{\text{oracle}}(x)$; fine-tuning set size $r$
2: **Output:** Unlearned model $\theta$
3: **for** $t = \{1, ..., T\}$ **do**                                                      ▷ $T$ epochs
4:      $S'_R \leftarrow S \setminus S_F$                                  ▷ Sub-sample $r$ points from retain set
5:      $S_{\text{fine-tune}} = S_F \bigcup S'_R$
6:      **for** $x \sim S_{\text{fine-tune}}$ **do**                                         ▷ mini-batch
7:          $L(\theta_t) = \|f_x(\theta_t) - f^{\text{oracle}}(x)\|^2$                   ▷ Compute loss
8:          $\theta_{t+1} = \theta_t - \eta_t \cdot \nabla_\theta L(\theta_t)$            ▷ Perform update with gradient
9:      **end for**
10: **end for**
11: **Return** Model $\theta = \theta_T$

---

### C.2 DATAMODEL DIRECT

---

**Algorithm C.2** DM-DIRECT

---

1: **Input:** Trained model $\theta_0$; datamodels $\beta(x)$ for each $x \in S$; forget set $S_F$
2: **Output:** A predictor $h(\cdot) : S \mapsto \mathbb{R}^k$
3: $h(x) := f_x(\theta_0) - \sum\limits_{i \in S_F} \beta_i(x)$
4: **End**

---

### C.3 DATAMODEL MATCHING

---

**Algorithm C.3** Datamodel Matching (DMM)

---

1: **Input:** Trained model $\theta_0$; datamodels $\beta(\cdot)$; fine-tuning set size $r$
2: **Output:** Unlearned model $\theta$
3: $S'_R \leftarrow S \setminus S_F$                                         ▷ Sub-sample $r$ points from retain set
4: $S_{\text{fine-tune}} = S_F \bigcup S'_R$
5: $h \leftarrow$ DM-DIRECT$(\theta_0, \beta, S_f)$                        ▷ Simulate oracles with datamodels
6: **for** $t = \{1, ..., T\}$ **do**                                              ▷ $T$ epochs
7:      **for** $x \sim S_{\text{fine-tune}}$ **do**                                        ▷ mini-batch
8:          $L(\theta_t) = \|f_x(\theta_t) - h(x)\|^2$                       ▷ Compute loss
9:          $\theta_{t+1} = \theta_t - \eta_t \cdot \nabla_\theta L(\theta_t)$            ▷ Perform update with gradient
10:      **end for**
11: **end for**
12: **Return** Model $\theta = \theta_T$

---

# D  EXPERIMENTAL SETUP

## D.1  TRAINING SETUP

For CIFAR-10, we train ResNet-9 models[5] for 24 epochs with SGD with a batch size of 512, momentum 0.9, and weight decay $5e - 4$. We set learning rate initially at 0.4, and a single-peak cosine schedule peaking at the 5th epoch. We use a momentum of 0.9 and a weight decay of $5e - 4$.

For ImageNet Living17 (Santurkar et al., 2020), we train ResNet-18 models for 25 epochs using SGD with a batch size of 1024, momentum 0.9, and weight decay $5e - 4$. Label smoothing is set to 0.1.

## D.2  CONSTRUCTING FORGET SETS

We evaluate methods across various types and sizes of forget sets to test the robustness of unlearning. Our selection of unlearning scenarios span both random and non-random forgets of different sizes; that said, we view the non-random sets as practically more interesting. Compared to prior work, our target sets are harder to unlearn as we remove a small coherent subpopulation as opposed to an entire class.

On CIFAR-10, we use 9 different forget sets: sets 1,2,3 are random forget sets of sizes 10,100,1000 respectively; sets 4-9 correspond to semantically coherent subpopulations of examples (e.g., all dogs facing a similar direction) identified using clustering methods.

On ImageNet Living-17, we use three different forget sets: set 1 is random of size 500; sets 2 and 3 correspond to 200 examples from a certain subpopulation (corresponding to a single original ImageNet class) within the Living-17 superclass.

## D.3  DATAMODEL ESTIMATION

**Regression-based.** We re-train models on random 50% subsets of the full train dataset, and use between 1,000 and 20,000 models. We use the sparse linear regression based solvers from Ilyas et al. (2022) to estimate each datamodel vector. Though our main results are computed with 20,000 models, we find that using just 1,000 models suffice effective unlearning with DMM.

**TRAK.** We compute TRAK scores using 300 model checkpoints and 16328 projection dimensions using the code provided in Park et al. (2023).

## D.4  UNLEARNING BASELINES

Most unlearning algorithms are highly sensitive to the choice of forget set; thus, so for each of the unlearning algorithms we compare to, we evaluate over a grid of hyperparameters and report the best KLoM scores *for each* forget set. Below we describe each algorithm and indicate the respective hyperparameter grid.

**I. Full/partial Re-training**
Full retraining discards the original model and trains a new model from scratch using only the retain set $S_R$. By definition, this achieves perfect unlearning. As additional baselines to tradeoff time and unlearning accuracy, we re-train for different number of epochs (on the same learning rate schedule), but keep all the other hyperparameters the same as in Appendix D.1.

**II. Gradient Ascent (GA)**
Gradient ascent iteratively updates model parameters to increase loss on the forget set. Starting from a trained model $\theta$, the algorithm computes the gradient of the loss $\mathcal{L}$ with respect to the model parameters over examples $\mathcal{D}_{\text{forget}}$. These gradients are then used to iteratively adjust $\theta$ in the direction that increases $\mathcal{L}$.

**Hyperparameters:** For each forget set, we sweep over the following hyperparameters:

---

[5]https://github.com/wbaek/torchskeleton/blob/master/bin/dawnbench/cifar10.py

    1. Learning rate ($\eta$): $[10^{-5}, 10^{-4}, 10^{-3}, 10^{-2}]$.

    2. Total unlearning epochs: $[1, 3, 5, 7, 10]$.

### III. Gradient Descent (GD)

Gradient descent achieves unlearning by fine-tuning the original model on the retain set $S_R$. We note that in the convex setting this algorithm provides provable unlearning guarantees Neel et al. (2021).

**Hyperparameters:** For each forget set, we swept over the following hyperparameters:

    1. Learning rate ($\eta$): $[10^{-5}, 10^{-4}, 10^{-3}, 10^{-2}]$.

    2. Total unlearning epochs: $[1, 3, 5, 7, 10]$.

### IV. SCRUB

SCRUB (Kurmanji et al., 2023), the current state-of-the-art unlearning method, combines variants of GA and GD-based heuristics. During the first phase, it runs a GA-like algorithm on the forget set while simultaneously running GD-like on the retain set; in the second phase, it runs additional iterations of GD on the retain set. For the first phase, SCRUB specifically maximizes/minimizes the KL divergence between the logits of the original model's predictions and the new unlearned model's predictions. See the original paper for more details.

**Hyperparameters:**

For each forget set, we sweep over the following hyperparameters:

    1. Momentum parameter $\beta$ was set to 0.999

    2. Retain batch size: 64

    3. Forget batch size: 32, 64

    4. Number of epochs for GA-based unlearning (maximization epochs) : 1,3,5,7,9

    5. Learning rates: $[3 \times 10^{-1}, 10^{-1}, 3 \times 10^{-2}, 10^{-2}, 3 \times 10^{-3}, 10^{-3}, 3 \times 10^{-4}, 10^{-4}, 3 \times 10^{-5}, 10^{-5}, 3 \times 10^{-6}, 10^{-6}]$.

    6. Number of total epochs: $[5, 7, 10]$

### V. Do-nothing

This is a "no-op" that simply returns the original (fully-trained) model. We include this as a baseline as: i) typically this is what is currently done in practice and ii) many existing methods (as our evaluations show) perform worse than this trivial baseline.

Implementations for all methods are available at:
bit.ly/unlearning-via-simulated-oracles

# E  LINEAR MODEL ANALYSIS

## E.1  PROOF OF THEOREM 1

We restate Theorem 1 below.

**Theorem 1** (Proof in Appendix E.1). *Let $S$ and $S_R$ be the full training set and the retain set respectively, with input matrices $X$ and $X_R$ and corresponding labels $y$ and $y_R$. Additionally, let $\theta_{full}$ and $\theta_*$ denote the optima of the ridge objective (5) for the full data $S$ and retain set $S_R$ respectively. After $t$ iterations of unlearning starting from $\theta_{full}$, the iterate $\theta_t$ satisfies*

$$\theta_t - \theta_* = \begin{cases} (I - 2\eta\lambda)^t \left(I - \frac{2\eta}{1-2\eta\lambda}X_R^\top X_R\right)^t (\theta_{full} - \theta_*) & \text{for ridge gradient descent (GD).} \\ \left(I - 2\eta X^\top X\right)^t (\theta_{full} - \theta_*) & \text{for oracle matching (OM).} \end{cases}$$

*Proof.* Note that by using the Taylor expansion of the ridge gradient descent objective around $\theta_*$, we can rewrite it as follows:

$$\|X_R\theta - y_R\|_2^2 + \lambda\|\theta\|^2 \tag{7}$$

$$= \|X_R\theta_* - y_R\|^2 + \lambda\|\theta_*\|_2^2 + (\theta - \theta_*)^\top \left(X_R^\top X_R + \lambda I\right)(\theta - \theta_*) \tag{8}$$

$$= c + (\theta - \theta_*)^\top \left(X_R^\top X_R + \lambda I\right)(\theta - \theta_*), \tag{9}$$

where $c = \|X_R\theta_* - y_R\|^2 + \lambda\|\theta_*\|^2$ is a constant independent of $\theta$. Similarly, we can write oracle matching objective as

$$\|X\theta - X\theta_*\|^2 = (\theta - \theta_*)^\top \left(X^\top X\right)(\theta - \theta_*). \tag{10}$$

Note that both the ridge gradient descent and the oracle matching objectives can be written as

$$(\theta - \theta_*)^T \left(Z^T Z\right)(\theta - \theta_*) + c \tag{11}$$

for some PSD matrix $Z^T Z$ and some constant $c$. Gradient descent update on objective 11 can be written as:

$$\theta_t = \theta_{t-1} - 2\eta \, Z^\top Z(\theta_{t-1} - \theta_*). \tag{12}$$

Subtracting $\theta_*$ from both sides,

$$\theta_t - \theta_* = (I - 2\eta Z^\top Z)(\theta_{t-1} - \theta_*). \tag{13}$$

Unrolling the recursion, squaring both sides, and simplifying then yields the desired result.  □

## E.2  PROOF OF THEOREM 2

**Theorem 2** (Proof in Appendix E.2). *Consider the setting of Theorem 1, where $\theta_t$ is the iterate after $t$ steps of unlearning initialized at $\theta_{full}$. Further let $\gamma_{min}, \gamma_{max}$ denote the minimum and maximum eigenvalues of $X^\top X$, and let $A$ and $B$ be lower and upper bounds on the norm of the covariates (i.e., $B \geq \|x_i\| \geq A$ for all $i$). Then, as long as the learning rate $\eta \in (0, \frac{2}{5(\gamma_{max}+\lambda)})$ and $\text{rank}(X) > 1$,*

$$\frac{\mathbb{E}\left[\|\theta_t - \theta^*\|^2\right]}{\|\theta_{full} - \theta_*\|^2} \in \begin{cases} O\left((1 - \gamma_{min}\eta)^t\right) & \text{for oracle matching (OM).} \\ \Omega\left(\frac{1}{t^2}\right) & \text{for ridge gradient descent (GD).} \end{cases} \tag{6}$$

*Proof.* We will begin with a more general setting than the theorem. In particular, consider an arbitrary convex optimization problem of the form

$$\min_\theta f(\theta), \qquad \text{where } f(\theta) = \frac{1}{n}\sum_{i=1}^n f_i(\theta),$$

where the function $f$ is $\alpha$-strongly convex, and each $f_i$ is $\beta$-smooth. In other words, we have that for any $\theta$ and $\theta'$,

$$\langle \nabla f(\theta) - \nabla f(\theta'), \theta - \theta'\rangle \geq \alpha\|\theta - \theta'\| \qquad (f \text{ is } \alpha\text{-strongly convex})$$

$$\|\nabla f_i(\theta) - \nabla f_i(\theta')\| \leq \beta\|\theta - \theta'\| \text{ for all } i \in [n]. \qquad (\text{each } f_i \text{ is } \beta\text{-smooth})$$

Note that without loss of generality, we can restrict our attention to the subspace spanned by $X$, in which case both oracle matching and ridge gradient descent are instances of this setting—for GD, $\alpha = \gamma_{min} + \lambda$ and $\beta = \gamma_{max} + \lambda$, and for OM $\alpha = \gamma_{min}$ and $\beta = \gamma_{max}$.

We further define a quantity $\sigma_f^2$ called *gradient disagreement*, measured as

$$\sigma_f^2 := \mathbb{E}_i \left[ \|\nabla f_i(\theta_*) - \mathbb{E}[\nabla f_i(\theta_*)]\|^2 \right] = \mathbb{E}_i \left[ \|\nabla f_i(\theta_*)\|^2 \right],$$

where $\theta_*$ is the optimum of $f$.

**Upper bound for OM.** With these quantities defined, let us begin with the OM upper bound. In fact, this follows directly from a standard SGD convergence proof, e.g., Theorem 5.8 of (Garrigos & Gower, 2023), restated below:

**Theorem 3** (Theorem 5.8 of (Garrigos & Gower, 2023))**.** *Suppose $f$ is a $\alpha$-strongly convex sum of $\beta$-smooth convex functions. Consider the sequence of iterates $\{\theta_t\}_{t \in \mathbb{N}}$ generated by stochastic gradient descent with a fixed step size $\eta \in (0, \frac{1}{2\beta})$. For $t \geq 0$,*

$$\mathbb{E} \left[ \|\theta_t - \theta_*\|^2 \right] \leq (1 - \eta\alpha)^t \|\theta_0 - \theta_*\|^2 + \frac{2\eta}{\alpha} \cdot \sigma_f^2.$$

A few observations conclude the proof. First, any step size $\eta \leq \frac{2}{5(\gamma_{max}+\lambda)}$ also satisfies $\eta \leq \frac{1}{2\gamma_{max}}$ and we can thus apply the Theorem to our case. Second, for oracle matching, we have that

$$\nabla f_i(\theta) = 2 \left( \boldsymbol{x}_i^\top \theta - \boldsymbol{x}_i^\top \theta_* \right) \boldsymbol{x}_i,$$

which means that $\nabla f_i(\theta_*) = 0$ and thus $\sigma_f^2 = 0$, concluding the proof.

**Lower bound for GD.** We now show that for the same set of learning rates, the stochastic version of ridge gradient descent on the retain set cannot converge faster than $1/t^2$. Key to our analysis will be that, for GD, the gradient disagreement $\sigma_f^2$ is non-zero, so long as the dataset is non-degenerate. In particular,

$$\begin{aligned}
\sigma_f^2 &= \mathbb{E}_i \left[ \|\nabla f_i(\theta_*)\|^2 \right] \\
&\geq \frac{1}{n} \max_i \|\nabla f_i(\theta_*)\|^2 \\
&= \frac{4}{n} \max_i \left\| (\boldsymbol{x}_i^\top \theta_* - y_i)\boldsymbol{x}_i + \lambda\theta_* \right\|^2.
\end{aligned}$$

To see that this is strictly positive, we can proceed by contradiction. Suppose that $\sigma_f^2 = 0$. Observe that if $\boldsymbol{x}_i^\top \theta_* = y_i$ for any $i \in [n]$, then the corresponding $\|\nabla f_i(\theta_*)\|^2 = 4\lambda^2\|\theta_*\|^2$, and so $\sigma_f^2 > 0$. Thus, $\boldsymbol{x}_i^\top \theta_* \neq y_i$ for all $i$. In this case, however, we must have that

$$(\boldsymbol{x}_i^\top \theta_* - y_i)\boldsymbol{x}_i = -\lambda\theta_*,$$

meaning that $\theta_*$ is parallel to $\boldsymbol{x}_i$. If $\text{rank}(X) > 1$, this is a contradiction and so $\sigma_f^2 > 0$.

We can now continue with the rest of the proof. We start with some algebraic manipulation of the gradient update. In particular, for a random $i \in [n]$,

$$\begin{aligned}
\theta_t - \theta_* &= \theta_{t-1} - \theta_* - \eta\nabla f_i(\theta_{t-1}) \\
\|\theta_t - \theta_*\|^2 &= \|\theta_{t-1} - \theta_*\|^2 - 2\eta\langle\theta_{t-1} - \theta_*, \nabla f_i(\theta_{t-1})\rangle + \eta^2\|\nabla f_i(\theta_{t-1})\|^2.
\end{aligned}$$

Taking an expectation conditioned on $\theta_{t-1}$,

$$\mathbb{E} \left[ \|\theta_t - \theta_*\|^2 \right] = \|\theta_{t-1} - \theta_*\|^2 - 2\eta\langle\theta_{t-1} - \theta_*, \nabla f(\theta_{t-1})\rangle + \eta^2\mathbb{E} \left[ \|\nabla f_i(\theta_{t-1})\|^2 \right].$$

Now, we treat the second and third terms separately. In particular, for the second term,

$$\begin{aligned}
2\eta\langle\theta_{t-1} - \theta_*, \nabla f(\theta_{t-1})\rangle &\leq 2\eta\|\theta_{t-1} - \theta_*\|\|\nabla f(\theta_{t-1})\| && \text{(Cauchy-Schwarz)} \\
&= 2\eta\|\theta_{t-1} - \theta_*\|\|\nabla f(\theta_{t-1}) - \nabla f(\theta_*)\| && \text{(Gradient at optimum is zero)} \\
&\leq 2\eta\beta\|\theta_{t-1} - \theta_*\|^2. && \text{(Smoothness)}
\end{aligned}$$

For the third term, we use the identity $\|u - v\|^2 \leq 2\|u\|^2 + 2\|v\|^2$, which we can rearrange to be $\|u\|^2 \geq \frac{1}{2}\|u - v\|^2 - \|v\|^2$. Letting $u = \nabla f_i(\theta_{t-1})$ and $v = \nabla f_i(\theta_{t-1}) - \nabla f_i(\theta_*)$,

$$
\begin{aligned}
\eta^2 \mathbb{E}\left[\|\nabla f_i(\theta_{t-1})\|^2\right] &\geq \frac{\eta^2}{2}\mathbb{E}\left[\|\nabla f_i(\theta_*)\|^2\right] - \eta^2\mathbb{E}\left[\|\nabla f_i(\theta_{t-1}) - \nabla f_i(\theta_*)\|^2\right] \\
&\geq \frac{\eta^2}{2}\mathbb{E}\left[\|\nabla f_i(\theta_*)\|^2\right] - \eta^2\beta^2\|\theta_{t-1} - \theta_*\|^2 \\
&= \frac{\eta^2\sigma_f^2}{2} - \eta^2\beta^2\|\theta_{t-1} - \theta_*\|^2.
\end{aligned}
$$

Combining everything so far,

$$
\mathbb{E}\left[\|\theta_t - \theta_*\|^2\right] \geq \left(1 - 2\eta\beta - \eta^2\beta^2\right)\|\theta_{t-1} - \theta_*\|^2 + \frac{\eta^2\sigma_f^2}{2}
$$

Taking an expectation with respect to previous iterates yields

$$
\begin{aligned}
\mathbb{E}\left[\|\theta_t - \theta_*\|^2\right] &\geq \left(1 - 2\eta\beta - \eta^2\beta^2\right)^t\|\theta_0 - \theta_*\|^2 + \frac{\eta^2\sigma_f^2}{2}\cdot\sum_{\tau=0}^{t-1}\left(1 - 2\eta\beta - \eta^2\beta^2\right)^\tau \\
&\geq \left(1 - 2\eta\beta - \eta^2\beta^2\right)^t\|\theta_0 - \theta_*\|^2 + \frac{\eta^2\sigma_f^2}{2} \\
&\geq \left(1 - \frac{12}{5}\eta\beta\right)^t\|\theta_0 - \theta_*\|^2 + \frac{\eta^2\sigma_f^2}{2} \qquad\qquad \text{since } \eta \leq \frac{2}{5\beta}.
\end{aligned}
$$

For ease of notation, let $C = \frac{12}{5}\beta$. Note that $\log(1 - x) \geq \frac{x^2 - 2x}{1-x}$ for $x < 1$, and so

$$
\mathbb{E}\left[\|\theta_t - \theta_*\|^2\right] \geq \exp\left(t\cdot(-C\eta)\frac{2 - C\eta}{1 - C\eta}\right)\|\theta_0 - \theta_*\|^2 + \frac{\eta^2\sigma_f^2}{2}. \tag{14}
$$

We now derive the learning rate $\eta$ that minimizes (14), and show that at this optimal learning rate (14) $= \Omega(1/t^2)$. Note we can see this by inspection even without the formal derivation, because for (14) to be $O(\frac{1}{t^2})$ we need $\eta = O(\frac{1}{t})$ so that the right hand term is $O(1/t^2)$, which forces the first term $\exp\left(t\cdot(-C\eta)\frac{2-C\eta}{1-C\eta}\right) = O(1)$. Now, to minimize the right hand side above with respect to $\eta$, we take the derivative and set to zero (note that at the extreme points $\eta = 0$ and $\eta = \frac{2}{5\beta}$ we are left with a constant amount of error). The result of this calculation is the fixed learning rate that optimizes the error at time $t$. Again for ease of notation, let $g(\eta) = \frac{2-C\eta}{1-C\eta}$, so that

$$
\begin{aligned}
0 &= \frac{d}{d\eta}\left[\exp\left(t\cdot(-C\eta)g(\eta)\right)\|\theta_0 - \theta_*\|^2 + \frac{\eta^2\sigma_f^2}{2}\right] \\
&= -Ct\left(g(\eta) + \eta g'(\eta)\right)\exp\left(t\cdot(-C\eta)g(\eta)\right)\|\theta_0 - \theta_*\|^2 + \eta\sigma_f^2 \\
\eta\sigma_f^2 &= Ct\left(g(\eta) + \eta g'(\eta)\right)\exp\left(t\cdot(-C\eta)g(\eta)\right)\|\theta_0 - \theta_*\|^2 \\
\log(\eta\sigma_f^2) &= \log(t) + \log(C(g(\eta) + \eta g'(\eta))\|\theta_0 - \theta_*\|) - Ct\eta\cdot g(\eta) \\
Ct\eta\cdot g(\eta) - \log(t) &= \log(C(g(\eta) + \eta g'(\eta))\|\theta_0 - \theta_*\|) - \log(\eta\sigma_f^2) \\
Ct\eta\cdot g(\eta) &\geq \log(C(g(\eta) + \eta g'(\eta))\|\theta_0 - \theta_*\|) - \log(\eta\sigma_f^2) \\
&\geq \log\left(\frac{C(g(\eta) + \eta g'(\eta))\|\theta_0 - \theta_*\|}{\sigma_f^2}\right) + \log(1/\eta) \\
t &\geq \frac{\log\left(\frac{C(g(\eta) + \eta g'(\eta))\|\theta_0 - \theta_*\|}{\sigma_f^2}\right) + \log(1/\eta)}{C\eta\cdot g(\eta)}
\end{aligned}
$$

Now, by definition of $g(\eta)$, we have that for $\eta \in (0, \frac{2}{5\beta})$, $g(\eta) \le 26$ and $g(\eta) + \eta \cdot g'(\eta) \ge 2$. Thus:

$$t \ge \frac{\log\left(\frac{2C\|\theta_0 - \theta_*\|}{\sigma_f^2}\right) + \log(1/\eta)}{26 C \eta}$$

$$26 C t \ge \frac{\log\left(\frac{2C\|\theta_0 - \theta_*\|}{\sigma_f^2}\right) + \log(1/\eta)}{\eta}$$

Using the fact that $\log(1/x) \ge 1 - x$ for $x \in (0, 1)$ yields:

$$26 C t \ge \frac{\log\left(\frac{2C\|\theta_0 - \theta_*\|}{\sigma_f^2}\right) + 1}{\eta} - 1$$

$$\eta \ge \frac{\log\left(\frac{2C\|\theta_0 - \theta_*\|}{\sigma_f^2}\right) + 1}{26 C t + 1}.$$

Plugging this result into (14) yields the desired $\Omega(1/t^2)$ lower bound. $\qquad\square$

### E.3 EXPERIMENT DETAILS

**Details of unlearning algorithms**

We consider the various iterative algorithms for unlearning starting from $\theta_{\text{full}}$. For all of them, we consider their full-batch as well as the stochastic version. For the stochastic versions, we use a mini-batch size of 5. We search for the learning rate from $\{10, \frac{10}{2}, \frac{10}{2^2}, \cdots, \frac{10}{2^{20}}\}$. We describe the algorithms below:

1. **Ridge Gradient Descent (GD)**: This involves minimizing the ridge regression objective with the retain set points $(X_{\text{retain}}, y_{\text{retain}})$ using gradient descent.

2. **Ridge Gradient Descent + Ascent (GD+GA)**: This method aims to incorporate forget set points in the ridge gradient descent updates. Each step involves moving in a direction that is a linear combination of the gradient descent step on the retain set and the gradient ascent step on the forget set. That is, we set

$$\theta_t = \theta_{t-1} - \eta(\text{grad}_{\text{retain}}(\theta_{t-1}) - \alpha * \text{grad}_{\text{forget}}(\theta_{t-1})).$$

   Here, $\text{grad}_{\text{retain}}(\theta) = 2 X_{\text{retain}}^T (X_{\text{retain}} \theta - y_{\text{retain}}) + 2\lambda\theta$ and $\text{grad}_{\text{forget}}(\theta) = 2 X_{\text{forget}}^T (X_{\text{forget}} \theta - y_{\text{forget}})$. We do a hyperparameter search for $\alpha$ in $\{0.01, 0.1, 1, 10\}$.
   In the stochastic setting, in each update step, we draw minibatch points uniformly at random from the full dataset, and calculate the ascent step term only if the drawn points include points from the forget set.

3. **Oracle Matching (OM)**: Here, we assume oracle access to predictions made using the optimal model $\theta_*$. This method involves using gradient descent to minimize the the squared error from the oracle predictions on the full dataset: $\|X_{\text{full}}\theta - X_{\text{full}}\theta_*\|_2^2$. We include a full algorithm of this in Algorithm C.1.

4. **Oracle Matching on retain set (OM retain set)**: This involves using gradient descent to minimize the squared error from the oracle predictions only on the retain set points: $\|X_{\text{retain}}\theta - X_{\text{retain}}\theta_*\|_2^2$.

**Slow convergence with stochastic gradient descent.**

In Section 5, we saw that in the stochastic setting, OM converges much faster than GD, even when $\lambda$ is large. Here, we dig deeper into the large $\lambda$ experiment considered in Section 5 to understand this. We observe that stochastic GD remains stable only at small learning rates with large $\lambda$, which results in slower progress. Specifically, while the optimal learning rate for full-batch GD is similar in both large and small $\lambda$ regimes, for stochastic GD, it is about 100 times smaller in the large $\lambda$ regime. Using a higher learning rate for stochastic GD in the large $\lambda$ regime leads to instability and non-convergence, an issue not seen with stochastic OM.

In each update step, stochastic OM adjusts the model parameters only within the span of the random subset of points used for that update. On the other hand, stochastic GD decays the model parameters in other directions due to the regularization term. Although we want the parameters to decay in the subspace orthogonal to the span of the retain set, stochastic GD also decays the parameters in directions spanned by the retain set points that are not part of the current update set. As a result, using a high learning rate for stochastic GD in the large $\lambda$ regime disrupts parameters in the span of the retain set. Therefore, while increasing $\lambda$ improves full-batch GD's convergence speed and could potentially make it faster than OM, this advantage does not apply to stochastic GD, which has to use a much smaller learning rate.

In Figure E.1a, we illustrate how stochastic GD at high learning rates disrupts the model parameters within the span of the retain set. We plot the progression of relative squared distance to the optimal unlearned parameter within the span of retain set points, $\frac{||P_{\text{retain}}(\theta_t - \theta)||_2^2}{||\theta_{\text{full}} - \theta_*||_2^2}$. Here $\theta_t$ is the iterate at time $t$, $\theta_{\text{full}}$ is the model trained on the full dataset, $\theta_*$ is the optimal unlearned model, and $P_{\text{retain}}$ is the projection matrix onto the retain set span. We show this for iterates with the optimal learning rate, as well as for learning rates 4 times faster and 4 times slower. As the learning rate increases, the iterates tend to diverge in the span of the retain set.

In Figure E.1b, we plot the progression of relative squared distance to the optimal unlearned parameter orthogonal in the subspace orthogonal to the retain set points, $\frac{||P_{\text{orth-retain}}(\theta_t - \theta_*)||_2^2}{||\theta_{\text{full}} - \theta_*||_2^2}$, where $P_{\text{orth-retain}}$ is the projection matrix for the orthogonal subspace. Here, we observe that increasing the learning rate beyond the optimal rate leads to faster convergence. Thus, while increasing the learning rate beyond the optimal value accelerates convergence in the subspace orthogonal to the retain points, it harms progress in the span of the retain set points. Therefore, in the large $\lambda$ regime, stochastic GD must operate at small learning rates to avoid disrupting the model parameters in the span of the retain set.

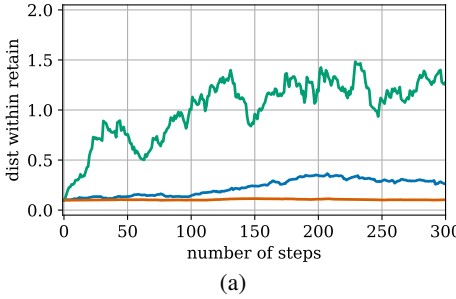 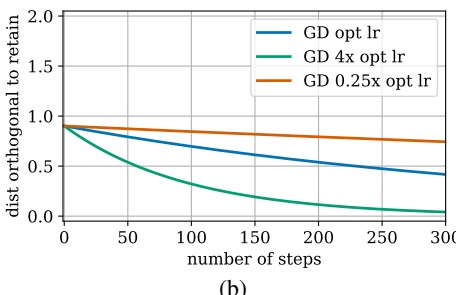

|  |
|---|
| (a) |

|  |
|---|
| (b) |

Figure E.1: Stochastic Gradient Descent performance in the large $\lambda$ regime with varying learning rates. (a) The y-axis shows the relative squared distance to the optimal unlearned parameter within the span of retain set points, $\frac{||P_{\text{retain}}(\theta_t - \theta)||_2^2}{||\theta_{\text{full}} - \theta_*||_2^2}$, where $\theta_t$ is the iterate at time $t$, $\theta_{\text{full}}$ is the model trained on the full dataset, $\theta_*$ is the optimal unlearned model, and $P_{\text{retain}}$ is the projection matrix onto the retain set span. Larger learning rates lead to divergence within this span. (b) The y-axis shows the relative squared distance to the optimal unlearned parameter in the subspace orthogonal to the retain set points, $\frac{||P_{\text{orth-retain}}(\theta_t - \theta_*)||_2^2}{||\theta_{\text{full}} - \theta_*||_2^2}$, where $P_{\text{orth-retain}}$ is the projection matrix for the orthogonal subspace. Larger learning rates result in faster convergence in this subspace.

## E.4 Additional Experiments

In section 5, we discussed an example with linear models that highlighted the qualitative differences between oracle matching and other unlearning methods. Here, we show the comparison for another example with different covariance structure. We draw 100 training points $(x_i, y_i)$ where $x_i$ are drawn i.i.d. from $N(0, \Sigma)$ in 400 dimensions where covariance matrix $\Sigma = diag(1, 1/2, 1/3, \cdots, 1/400)$ ( $diag(.)$ represents a diagonal matrix with the specified entries on the diagonal). $y_i = \theta^T x_i + \epsilon_i$, where $\theta$ is drawn from $N(0, I)$ and $\epsilon_i$ is drawn from $N(0, 1/4)$. These 100 points form $(X_{\text{full}}, y_{\text{full}})$. We fit these points to minimize the ridge regression objective with $\lambda = 1/4$ (for the small $\lambda$ case) or $\lambda = 5$ (for the large $\lambda$ case), to obtain the model $\theta_{\text{full}}$. Here $\lambda = 1/4$ is the $\lambda$ value that minimizes the

expected squared prediction error. We want to unlearn 5 training points chosen uniformly at random. We show the performance of various methods (in both the stochastic and full-batch setting with small and large $\lambda$) in Figure E.2. Even here, we obtain the same qualitative patterns as in Figure E.2.

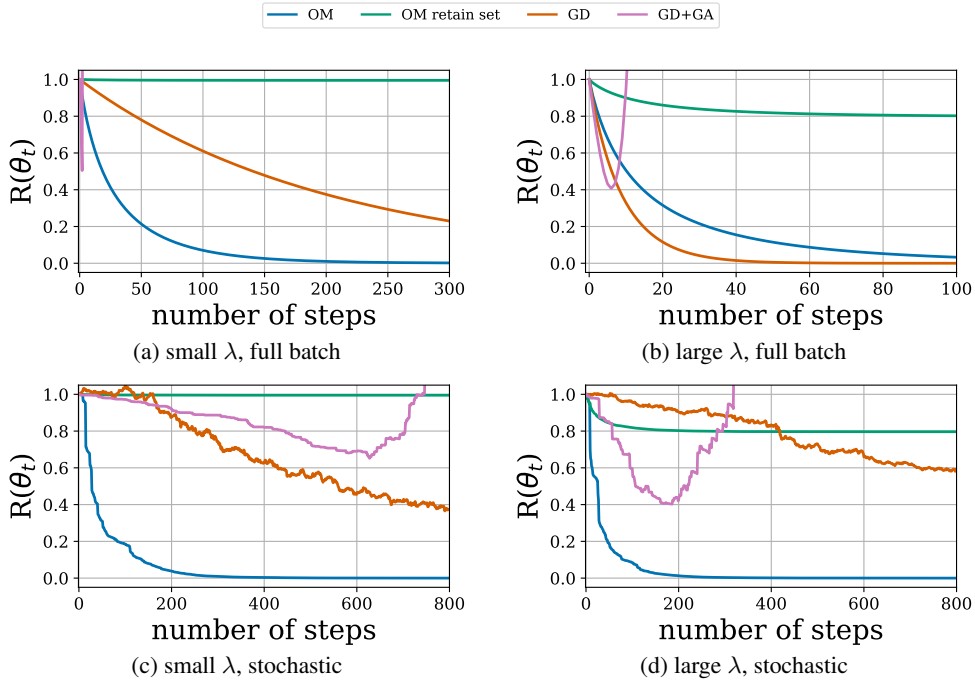

(a) small $\lambda$, full batch

(b) large $\lambda$, full batch

(c) small $\lambda$, stochastic

(d) large $\lambda$, stochastic

Figure E.2: Another example comparing the performance of various unlearning methods with linear models.

# F UNLEARNING EVALUATION

## F.1 KL DIVERGENCE OF MARGINS (KLoM)

Below we formally define the KLoM evaluation, which computes the distance between the distribution of outputs for unlearned models and re-trained models. For output, we use the classification margin. We also include a visual representation of our algorithm in Figure F.1.

---

**Algorithm F.1** KLoM

1: **Input** Number of models $N$, dataset $D$, forget set $F \subseteq D$, retain set $R \subseteq D$ (such that $D = F \cup R$), and a validation dataset $V$, training algorithm $A$, unlearning algorithm $U$, margin function $\phi$, and histogram function $H(S)$ .
2: Train $N$ models (*Oracles*) on the entire dataset, excluding the forget set. $\Theta^o = \{A(D \setminus F)\}$, $|\Theta^o| = N$.
3: Train $N$ models (*Unlearned-models*) on the entire dataset, then for each model unlearn forget set $F$, $\Theta^f = \{U(A(D), F)\}$, $|\Theta^f| = N$.
4: Initialize a vector of results with all zeros $\vec{r}$.
5: **for** each point $x$ in {Forget, Retain, Validation} set **do**
6:     Compute the margins for each oracle $M_o = \{\phi(\theta_i^o(x))|\theta_i^o \in \Theta^o\}$.
7:     Compute the margins for each unlearned-model $M_f = \{\phi(\theta_i^f(x))|\theta_i^f \in \Theta^f\}$
8:     Assign $\vec{r}[x] = KL(Hist(M_o), Hist(M_f))$
9: **end for**
10: return $\vec{r}$

---

Note that in order to approximate the KL divergence, we compute a histogram $H(S)$ that takes a set of real numbers and returns an empirical probability distribution by truncating and binning samples from $S$.

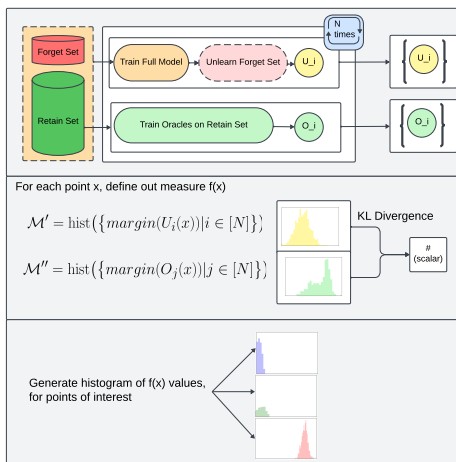

Figure F.1: Visual diagram of KLoM

In practice, we compute KLoM using $N = 100$ oracles compared to $N = 100$ unlearned models. For each point we evaluate we clip the $N$ margins to a range of $[-100, 100]$ to exclude outliers (some methods, like SCRUB, result in extremely large margins) into 20 bins. Then the KL-divergence is computed between the binned histogram of the oracles and the binned histogram of the unlearned models. For any region of the support where one histogram has no support, and the other does, the bin with no support is set to a non-zero probability of $\epsilon = 10^{-5}$. We note, that for this reason, KLoM scores are artificially rescaled and capped at value around $\approx 12$. This cap can be changed by changing the number of bins or the minimum value $\epsilon$.

## F.2 `U-LiRA`

At a high-level `U-LiRA` measures the distinguishability of predictions of an unlearned model from that of retrained models based on adapting membership inference attacks. Implementing the Algorithm F.2, as written, would be computationally infeasible in most cases, as it involves unlearning and retraining $T$ times for each point $(x, y)$. In practice, Hayes et al. (2024) computes `U-LiRA` using the method Algorithm F.4.

Omitting a few details, the main idea is as follows: consider an unlearning algorithm $\mathcal{U}$ and a training algorithm $\mathcal{A}$. Start from a model $\theta_0$ trained on a random subset $S \sim \mathcal{P}^n$, and unlearn one specific (random) forget set, to produce $\theta_F$. Next, construct a collection of shadow models by that producing many unlearned models from random training sets and random forget sets. Lastly for a collection of points in the retain set and in the forget set ($\{x | x \in S_F \cup S\}$) compare how $\theta_F$ compares to the subset of shadow models that never-saw-$x$ distribution of margins for models that unlearned-$x$. Now if $\mathcal{U}$ perfectly unlearns $S_F$, then $\theta_0$ no longer depends on $S_F$, and so even *conditioned* on $\theta_0$ the marginal distribution of $x \in S_F | \theta_0$ is still $\mathcal{P}$; the same distribution as any $x \in S_V$. Operationalizing this intuition, `U-LiRA` with probability $\frac{1}{2}$ draws either $x \in S_F$ or $x \in S_V$, and measures the output $y = f_x(\theta_0)$. An (optimal) adversary observes $y$, and tries to guess whether the corresponding $x$ was an unlearned point or a validation point, e.g. whether $x \in S_F$ or $S_V$. More generally, if $\mathcal{U}$ is an $(\epsilon, \delta)$-unlearning algorithm, the two distributions $y | x \in S_F, \theta_0$ and $y | x \in S_V, \theta_0$ would be $(\epsilon, \delta)$-indistinguishable by post-processing, and so even the optimal adversary couldn't have accuracy greater than $\frac{1}{2}e^\epsilon + \delta$. The optimal adversary can be implemented by training models with/without $x$, unlearning $S_F$, and then measuring the output $y$. For a more detailed description of `U-LiRA` and overview of similar MIA-based approaches, we refer the reader to (Hayes et al., 2024) (Section 4.2), and we include the pseudocode for the computationally efficient version of this evaluation Efficient-ULIRA in Appendix F.2.

---

**Algorithm F.2** U-LiRA (LiRA adapted for machine unlearning) (Hayes et al., 2024)

**Args:** model parameters to evaluate $\theta^*$, learning algorithm $A$, unlearning algorithm $U$, number of shadow models $T$, example $(x, y)$, logit function $\phi$, function that returns probabilities $f(\cdot, \theta)$ given model parameters $\theta$.
**Observations:** $O \leftarrow \{\}, \hat{O} \leftarrow \{\}$
**while** $t \leq T$ **do**

- $D \leftarrow$ sample a dataset that includes $(x, y)$
- $\theta^0 \leftarrow A(D)$ *train a model*
- $\theta' \leftarrow U(\theta^0, (x, y))$ *unlearn* $(x, y)$
- $\theta'' \leftarrow A(D \setminus (x, y))$ *retrain without* $(x, y)$
- $O[t] \leftarrow \phi(f(x; \theta'))$
- $\hat{O}[t] \leftarrow \phi(f(x; \theta''))$

**end while**
$\mu, \sigma \leftarrow$ *fit Gaussian($O$)*
$\hat{\mu}, \hat{\sigma} \leftarrow$ *fit Gaussian($\hat{O}$)*
$o \leftarrow \phi(f(x, \theta^*))$
$p_{\text{member}} \leftarrow \frac{N(o; \mu, \sigma^2)}{N(o; \mu, \sigma^2) + N(o; \hat{\mu}, \hat{\sigma}^2)}$
**if** $p_{\text{member}} \geq \frac{1}{2}$ **then**

- **return** Predict $(x, y)$ is a member of training

**else**

- **return** Predict $(x, y)$ is not a member of training

---

In the original `U-LiRA` paper (Hayes et al., 2024), they report results for Efficient `U-LiRA` for N= 256, forgettable points $F$ all points of class 5, $n_f = 40$ random forget sets per base model, and each forget set $m = 20$. The $N$ base models that they are train are trained on ResNet-18 for 100 epochs, and so are highly overparameterized.

---

**Algorithm F.3** Sub algorithm: Membership Prediction

---

1: **Input**: Sets $A$, $B$ of real-valued numbers, and point $x \in \mathbb{R}$.
2: $(\mu, \sigma \leftarrow$ fit Gaussian$(A)$
3: $(\hat{\mu}, \hat{\sigma} \leftarrow$ fit Gaussian$(B)$
4: $p_{\text{member}} \leftarrow \frac{\mathcal{N}(x; \mu, \sigma^2)}{\mathcal{N}(x; \mu, \sigma^2) + \mathcal{N}(x; \hat{\mu}, \hat{\sigma}^2)}$
5: **if** $p_{\text{member}} > \frac{1}{2}$ **then**
6:     **return** Predict $(x, y)$ is a *member* of set $A$
7: **else**
8:     **return** Predict $(x, y)$ is *not a member* of set $A$
9: **end if**

---

**Algorithm F.4** Efficient-ULIRA

---

1: **Input** Number of base models N, set of forgettable points $F$ (default: all points of class 5), Number of random forget sets per base model $n_f$, size of each forget set $m$ (default 200)
2: Train $N$ base models on random 50% subsets of the dataset
3: **for** each base model $\theta_i$ **do**
4:     construct $n_f$ random forget sets of size $m$, denoted $F_{i,j}$.
5:     **for** each random forget set $F_{ij}$ **do**
6:         Unlearn forget set $F_{ij}$
7:     **end for**
8: **end for**
9: Split the $n_f \cdot N$ unlearned models into two sets, Shadow models $S$ and Target models $T$
10: initialize accuracy vector $\vec{a} \in \mathbb{R}^{|T|}$, with all 0's.
11: **for** each target model $\theta_{\sqcup} \in T$ **do**
12:     Construct $D_f$ from $m$ from the $m$ points that $\theta_t$ unlearned
13:     Construct $D_v$ from $m$ from the $m$ points that $\theta_t$ was not trained on.
14:     let $D = D_f \cup D_v$
15:     Let $c = 0$
16:     **for** each point $x \in D$ **do**
17:         Let $S_A$ be the set of shadow models that were trained on $x$.
18:         Let $S_B$ be the set of shadow models that unlearned $x$
19:         Construct sets $A = \{\theta'(x) | \theta' \in S_A\}$, $B = \{\theta'(x) | \theta' \in S_B\}$
20:         Run sub-algorithm Membership Prediction F.3 with inputs $(A, B, x)$, returning $l \in \{1, 0\}$
21:         **if** ( **then**$l = 1$ and $x \in D_f$) OR ($l = 0$ and $x \in D_r$)
22:             $c+ = 1$
23:         **end if**
24:
25:     **end for**
26:     Average the model accuracy across all predictions in $D$, $a_t = c/(2m)$
27: **end for**
28: Average the accuracy-per-model over all the target models Return mean$(\vec{a})$

---

For our evaluation, `U-LiRA` paper ([Hayes et al., 2024](#)), we run Efficient `U-LiRA` for N= 50, $n_f = 40$ random forget sets per base model, and then we vary the training setting, the forget size $n_f$, and total set of forgettable points $F$. Specifically, we try 3 settings

1. ResNet-18 trained for 100 epochs, evaluated on forget sets of size 200, with the forgettable points $F$ being 1000 points in class 5.

2. ResNet-9 trained for 25 epochs, evaluated on forget sets of size 200, with the forgettable points $F$ being 1000 points in class 5.

3. ResNet-9 trained for 25 epochs, evaluated on forget sets of size 50, with the forgettable points $F$ being 500 random points in the dataset.

### F.3 COMPARING `U-LiRA` TO `KLoM`

In this paper we propose a new method for evaluating unlearning, `KLoM`. Here, we argue that this is a superior measure to existing measures, in particular compared to `U-LiRA`. The advantages of `KLoM` are :

1. `KLoM` requires fewer unlearning trials (on the order of 100) than `U-LiRA` (which is generally on the order of 2000).

2. `KLoM` returns a distribution of differences, rather than a binary assignment of if one particular model was more like an unlearned model or a retrained model. This is valuable because it tells you how a method unlearns individual points (e.g. is bad on average, or just bag on specific points)

3. `KLoM` does not assume the margins can be fit well by a Gaussian. Anecdotally, for `U-LiRA`, we find this is a decent but not great assumption, and it's currently unclear how much error this really introduces.

4. `KLoM` has the capacity to look at an unlearning algorithm's ability to handle coherent sets of points, not just random subsets of some set of forgettable points.

5. `U-LiRA` does not capture closeness to unlearned model, and thus one can force `U-LiRA` score to go down (implying better unlearning) by having the unlearning method destroy the original model, thus `U-LiRA` scores must be traded-off against an accuracy drop. `KLoM` measures distance directly, and thus unlearning can be evaluated with a single measure. We expand on this point below. And is illustrated by figure [F.2](#).

**A Toy Example Where `U-LiRA` Fails:**

Consider your unlearning method returns a constant function $f$ (e.g. such that that the margin is always some constant, e.g. 7). In such a situation, the distribution of unlearned models will look radically different from the distribution of models that were fully retrained; in theory, a good unlearning measure should return that this is a bad unlearning method. However, `U-LiRA` will actually return a nearly perfect unlearning score ($\approx 50\%$). In `U-LiRA`, one does 2 sets of likelihood ratio tests, first on points the unlearned model unlearned, then on points the unlearned model never saw. The first set of likelihood ratio tests, `U-LiRA` will get 100% (or nearly), because at margin is constant (7), and so it will be at the unlearned-models peak; thus we get 100% accuracy on these points. Now, on set 2, `U-LiRA` will get 0% (or nearly), because `U-LiRA` compute the likelihood ratio at the margin of the unlearned model's prediction, which in this case is a constant (7), thus it will still be at the unlearned-models peak. Thus, `U-LiRA` will not predict that anything is "not-in-the-training-set," achieving 0% on this set.

However, the problem is that 100 +0 /2 = 50%, which appears to be a perfect unlearning score, but in reality is just a classifier that thinks everything is from an unlearned model.

**Advantages of `U-LiRA`:**

1. It is worth noting that our measure requires binning, and some assumptions there. `U-LiRA` does not (because it makes the Gaussian approximation assumption)

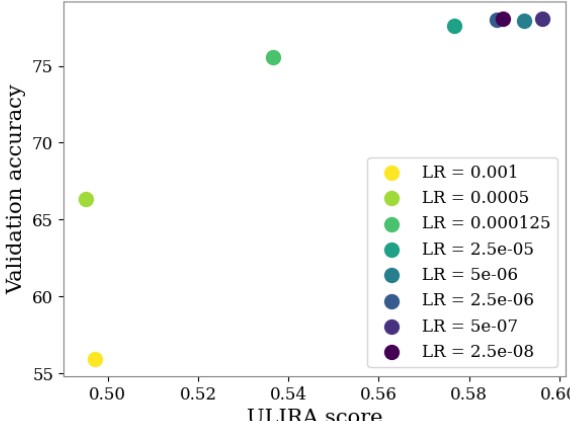

Figure F.2: `U-LiRA` accuracy vs Validation accuracy of the unlearned model, for Gradient Ascent method, for a model trained on CIFAR-10. Unlearning occurs on a random forget sets of size 200 in class 5. Observe that it is possible to achieve nearly perfect `U-LiRA` accuracy (50%) by simply dropping the validation accuracy of the model by 20%. With Gradient Ascent specifically, this is achieved by increasing the learning rate beyond the optimal learning rate, causing the gradient ascent updates to be too significant.

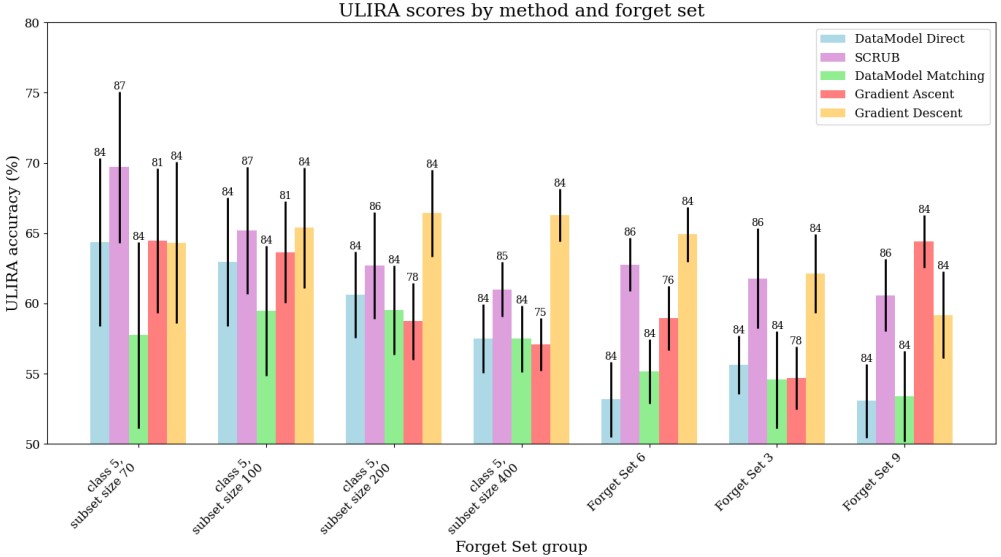

Figure F.3: Results for Efficient-ULIRA run with parameters specified in F.2. The validation accuracies are presented above the bar for the each method-forget set pairing. Observe that Gradient Ascent method is able to achieve an excellent `U-LiRA` score, but does so at the cost of the validation accuracy of the unlearned model.

2. `U-LiRA` reports an accuracy for 1 particular model, and then averages that across multiple unlearning models. this correctly captures individual model performance, rather than aggregate unlearning algorithm performance.

## F.4 SENSITIVITY OF UNLEARNING TO MODELS AND FORGET SETS

See Figure F.3.

# G UNDERSTANDING EFFECTIVENESS OF DATAMODEL MATCHING

We saw that datamodel matching is an effective and efficient algorithm for unlearning. Here, we aim to better understand the effectiveness of datamodel matching (DMM). Since DMM consists of i) oracle matching (the finetuning algorithm) and ii) datamodels (approximation to oracle outputs), we study each component separately. First, in Appendix G.1, we analyze the stability of OM across time and to different choices of hyperparameters and show:

- **OM is stable across time**: We show that once OM unlearns an example, the example generally stays unlearned after further iterations, addressing the original "missing targets problem." As a result, OM is also much more stable than prior methods with respect to the choice of optimization hyperparameters.

- **OM generalizes from a small sample**: Though OM introduces additional design parameters (sampling ratios for forget and retain sets), we find that OM is effective as long as we include a sufficiently large sample of both. In particular, we only need to sample a small fraction of the retain set, making OM efficient.

Next, in Appendix G.2, we ablate different components of the datamodel estimators to better understand necessary ingredients for DMM and show:

- **Necessity of modeling interactions between datapoints**: Datamodels (linearly) model the effect of different training examples on other inputs. We show that using only the "diagonal" entries (i.e., modeling only the self-influence) is much less effective.

- **Effectiveness of fast approximate data attribution methods**: We show that replacing regression-estimated datamodels with much faster alternatives like TRAK still yield effective unlearning algorithms, albeit with worse performance than well-estimated datamodels.

## G.1 UNLEARNING STABILITY OF OM

To motivate our approach, we demonstrated earlier that existing fine-tuning approaches suffer from the problem of different unlearning rates due to lacking meaningful targets to converge to. In Figure G.1, we find that OM no longer suffers from the same problem, since unlearning quality generally only improves over time (there is no risk of overshooting), even if points are still unlearned at different rates. Because of this, we find that OM is much more robust to the choice of optimization parameters such as learning rate and number of epochs compared to prior gradient ascent based methods (see Appendix H.1).

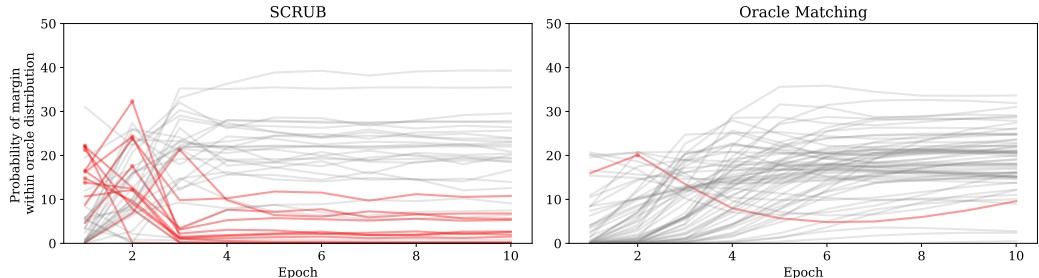

Figure G.1: **Oracle matching circumvents the stopping time problem.** We revisit the earlier plot and apply the same analysis to oracle matching. The red lines highlight examples in the forget set whose unlearning quality is hurt by training longer. This happens frequently when running SCRUB, but goes away nearly entirely when using oracle matching.

**Generalization of OM.** OM fine-tunes on samples from both the forget and retain sets. The efficiency of OM hinges on whether it can generalize from a small sample. Ablations (Appendix G.1) show i) OM succeeds as long as the ratio of forget set points in the fine-tuning set is sufficiently high, and ii) a small fraction ($\geq 0.04$) of retain set suffices to guide OM to converge towards an oracle on most

retain set samples. That is, OM is able to effectively *generalize* from a small sample of oracle outputs. This implies that we only need to approximate oracle predictions on a small fraction of the training data, increasing the efficiency of the OM algorithm.

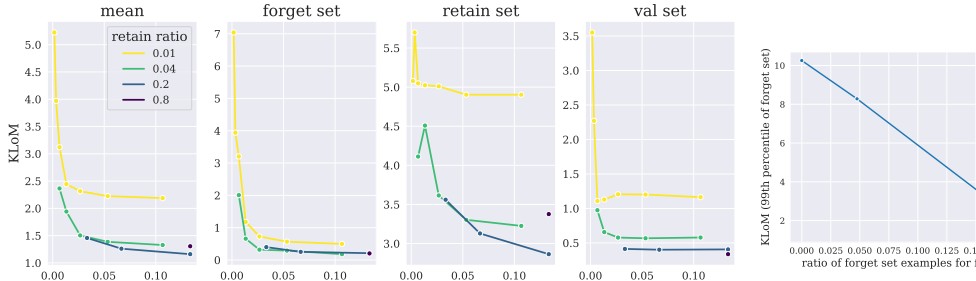

Figure G.2: Varying the fraction of retain set sampled for oracle matching. A sufficiently large sample ($\geq 0.04$) appears to be sufficient in enabling OM to generalize to out-of-sample.

Figure G.3: Varying the ratio of forget set samples in the fine-tuning set for oracle matching.

**Challenge of overfitting.**

We hypothesize that OM struggles on the retain set of Living-17 due to models being more overfit. A further investigation showed the lack of use of data augmentation as the main cause of overfitting.[6] Under new hyperparameter settings that are identical except the use of standard image data augmentation (random cropping and flips), the trained models overfit much less. In this new setting, we re-trained oracle models and applied oracle matching with the same exact hyperparameters as before (so they are not optimized for the new setup). The resulting KLoM scores on the retain set are significantly lower (95th percentile decreasing from 2.88 to 0.62; see Appendix G.1), supporting our hypothesis that significant overfitting caused OM to perform worse on the retain set on Living-17.

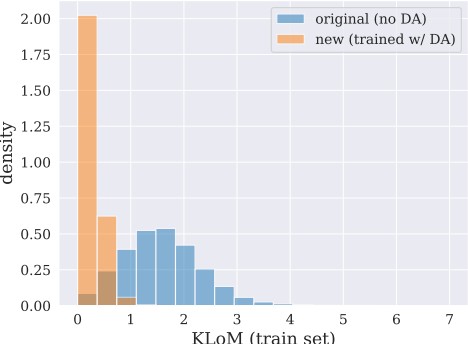

Figure G.4: **Oracle matching struggles more on overfit predictions.** We contrast the performance of oracle matching on the train set in two different hyperparameters settings, differing only in their use of data augmentation (DA). OM struggles much more on models that are overfit when data augmentation is turned off.

## G.2 DATAMODEL ABLATIONS

Given the stability of OM, the success of DMM essentially only depends on the fidelity of datamodel approximation to oracle outputs. We now study which components of the datamodel are in fact necessary.

---

[6]We had turned off data augmentation for all of our experiments as this was done on all major prior works on (empirical) unlearning for DNNs.

**Necessity of modeling interactions between datapoints.** Datamodels model the effect of different training examples on a given model output. By leveraging them, we were able to accurately simulate the oracle model outputs. Inspecting the weights of linear datamodels show that the dominant entry for the datamodel of a training input corresponding the example itself (i.e., excluding that training example has the largest negative effect on the model prediction on itself). The corresponding weight is what is also known as the *memorization score* in prior work Feldman (2021). Could memorization scores suffice to linearly model oracle outputs? In Figure G.6, we evaluate the quality of oracle predictions when using the full datamodel vector vs. only the memorization scores, and find the following:

- **Insufficiency of memorization scores for unlearning non-random forget sets:** Using only the diagonal entries hurts unlearning quality for non-random forget sets (5), particularly for examples in the tail (dotted line). Intuitively, this is because the other weights in datamodels (the "non-diagonal" entries) capture important cross-example correlations (e.g., similar examples should have reinforcing effect on one another). Moreover, globally scaling the memorization scores (orange lines) cannot account for the missing non-diagonal entries. On the other hand, this is less important for random forget sets (3), as two random examples are in general unrelated to one another. Hence, to effectively unlearn forget sets that arise in practice via our approach, it seems necessary to model interactions between different datapoints.

- **Consistency of best scaling:** As an artifact, we also find that scaling down the datamodel weights globally by a factor ($\approx 0.9$ here) improves unlearning quality marginally. Fortunately, the scale seems consistent across different types of forget sets; one could calibrate this scale using a "held-out" forget as part of a pre-computation stage, and subsequently apply to all forget sets.

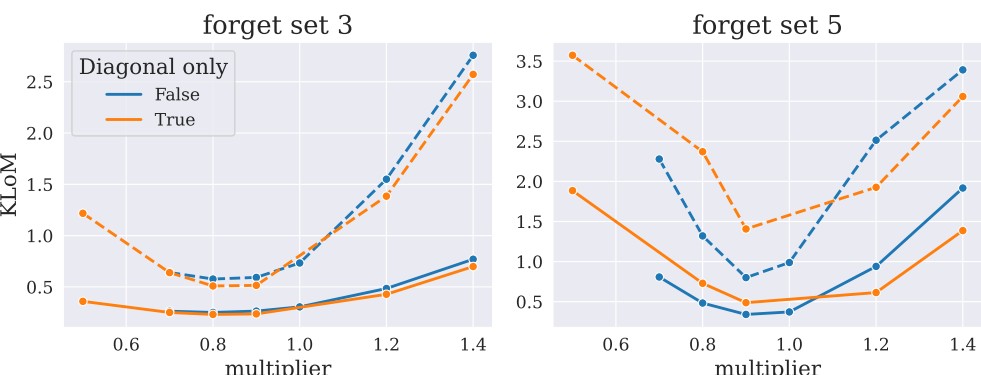

Figure G.5: The effect of non-diagonal entries and re-scaling on the unlearning effectiveness of DM-DIRECT for two different types of forget set on CIFAR-10 (left is random; right is non-random). Solid and dotted lines correspond to the mean and 95%-percentile KLoM scores. Diagonal-only indicates that we only use the memorization scores (the "diagonal" of the datamodel matrix).

**Scaling with estimation cost.** Though we excluded the cost of estimating datamodels in our analysis in Section 4.4 (as it is a one-time cost), practically we need to account for them as estimating predictive datamodels is computationally expensive. But estimating them is not all or nothing: we can tradeoff the computational cost and the datamodel predictiveness by varying the number of re-trained models. Here, we investigate how varying the computational resources affects both datamodel predictiveness (LDS) and unlearning performance (as measured by KLoM). In Figure G.6, we show the result of varying the computational cost by orders of magnitude: we can observe that while the datamodel predictiveness (as measured by the *linear datamodeling score* (Park et al., 2023)) continues increase at the same rate, KLoM—averaged across various forget sets—begins to saturate.

We hypothesize that this is due to the following difference in the distribution of counterfactual subsets being evaluated: while achieving high LDS requires predicting model outputs on the same distribution of random subsets of the training data that the datamodels were trained on, achieving high KLoM

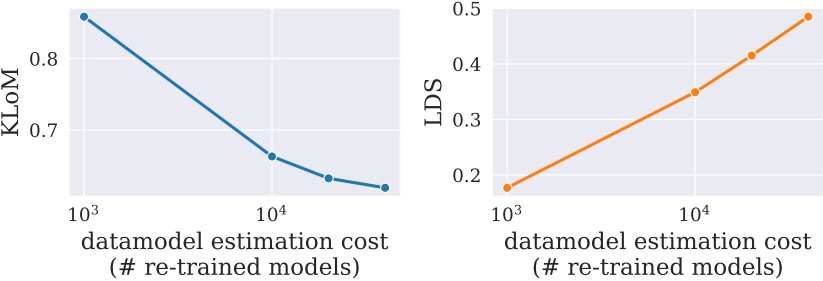

Figure G.6: **The effect of estimation cost on unlearning performance and datamodel predictiveness.** On CIFAR-10, we show how unlearning performance of DM-DIRECT (measured by KLoM; lower is better) and datamodel predictiveness (measured by the linear datamodeling score; higher is better) scales with datamodel estimation cost (number of re-trained models, in $\{10^3, 10^4, 2 \times 10^4, 4 \times 10^4\}$). KLoM is averaged over different forget sets.

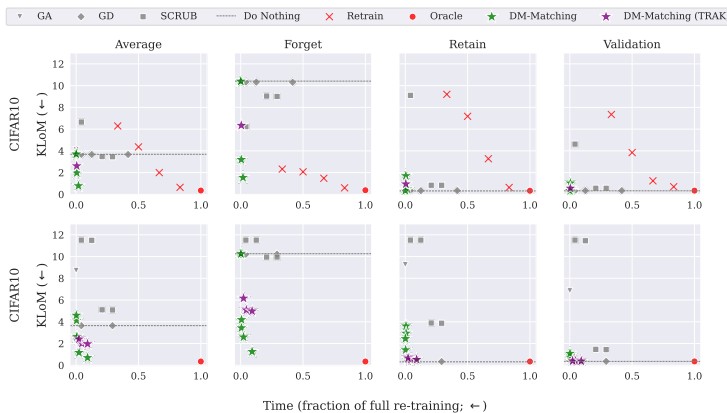

Figure G.7: **Datamodel-Matching with more efficient estimators.** In the same set up as in Figure 1, we evaluate the effectivness of DM-DIRECT and DMM when a different, more efficient estimator (TRAK) for datamodels. Though the equality of unlearning degrades, our algorithms still outperform prior methods. Note that the computational savings from using TRAK is not reflected on the x-axis, as computation of datamodels considered separately from the fine-tuning cost of DMM.

scores requires predicting over small non-random forget sets. In practice, this suggests that for the purposes of unlearning, cheaper alternatives (either by using the same regression-based estimator with reduced computational resources or by using a different estimator) may perform nearly as well as computationally expensive methods.

**Efficient unlearning with TRAK.** While datamodels estimated using the regression approach of Ilyas et al. (2022) are predictive, and can be pre-computed prior to unlearning, they nonetheless are expensive to compute. Since the work of Ilyas et al. (2022), follow-up works (Park et al., 2023; Grosse et al., 2023) have shown that efficient alternative methods can be quite effective with substantially lower computational costs.

Here we investigate whether DMM is still effective when datamodels are estimated with TRAK, which is based on a particular approximation to the influence function. In Figure G.7 we run OM with predictions generated by TRAK estimators with x1000 less compute than the regression-based datamodels. As expected DMM with TRAK performs worse in terms of KLoM than when datamodels are used, but we find that this drastically cheaper alternative to DMM still outperforms all prior methods significantly. These results highlight that improving the accuracy of TRAK in order to close the gap in KLoM scores with datamodels represents an important direction for developing even more efficient and effective unlearning algorithms.

# H  ADDITIONAL RESULTS

## H.1  HYPERPARAMETER SENSITIVITY OF UNLEARNING ALGORITHMS

Methods that involve something akin to gradient ascent, like SCRUB, where unlearning for longer or with a larger learning rate can reduce the utility of the model; as opposed to gradient descent based methods, like Oracle Matching, where if the learning rate is sufficiently small, unlearning for longer does not hurt the model. We illustrate this by showing how much the same unlearning algorithm is affected by changing the learning rate, in Figure H.1, where we plot the 85th percentile of KLoM score as a function of learning rate[7].

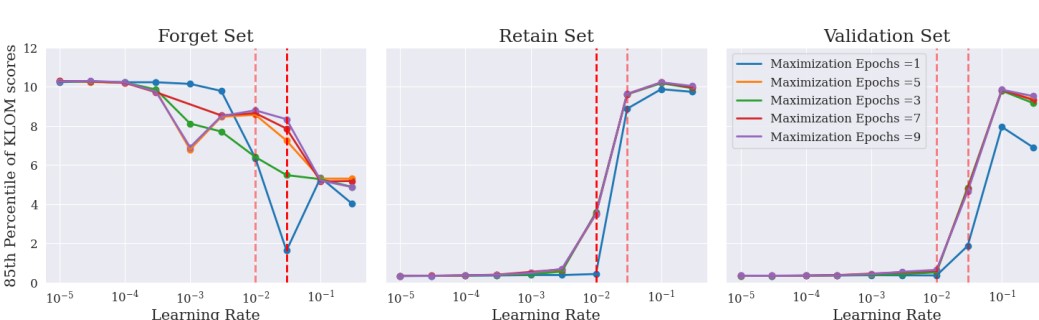

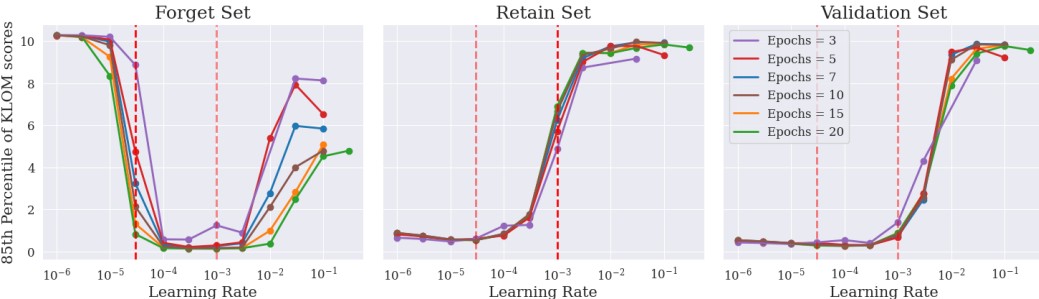

Figure H.1: KLOM sensitivity to learning rate. SCRUB is trained for 10 epochs, with the first "Maximization Epochs" used for Gradient Ascent; Oracle Matching is trained for 'Epochs', both demarcated in the legend. Both algorithms are evaluated on Forget Set 5, which is a non-random forget set of size 100. The red lines denotes the learning rate at which the KLOM score drops below 6 (an arbitrarily chosen threshold which we deemed as the largest value approaching a "reasonable" unlearning). Left red line indicates the largest forget that achieves a reasonable unlearning for Forget points, and the right red line indicates the smallest learning rate that achieves a reasonable unlearning for Validation and Retain points. The line is darker if that subset is the limiting factor for the thresholding.

The key takeaway is that the range of learning rates for SCRUB is much smaller (less than an order of magnitude), while Oracle Matching not only performs significantly better, the range of learning rates that attain good KLOM scores is much larger dynamic range. We do note that because this is a log-scale, the range for the SCRUB learning rates is larger in magnitude; however, the critical observation is how flat the KLOM scores are as a function of learning rate and the relative ranges of learning rates. Flatter and wider ranges (even on a log-scale) make hyperparameter tuning dramatically simpler and significantly more likely to choose a learning rate in a good range.

---

[7]We choose the 85th percentile for KLOM scores, because SCRUB generally performs significantly worse than Oracle Matching, and we found that for higher percentiles, the effect was weaker because SCRUB never achieved particularly good KLOM scores.

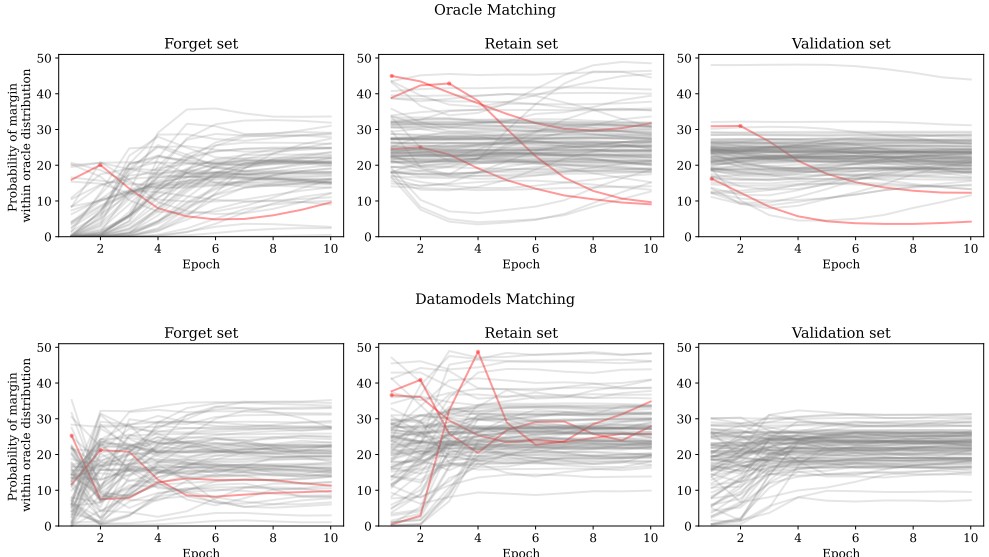

Figure H.2: Plotting unlearning quality over time for Oracle Matching and Datamodels matching. We observe that different points unlearn at different rates. However, unlike for gradient ascent based methods like SCRUB, for gradient descent based methods like Datamodels matching and oracle matching, one a data point is unlearned, it tends to remain unlearned. The red lines highlight examples whose final quality of unlearning is more than 10% worse than their maximum unlearning quality.

## H.2 PER-SAMPLE UNLEARNING OVER TIME

We present the rate of unlearning individual data points as a function of epochs in the unlearning algorithm for both Oracle Matching and Data Models matching, which we contrast with SCRUB unlearning, presented in Figure 2.

See Figure H.2.

## H.3  FULL KLoM EVALUATION

Building on this point that the effectiveness of different methods is forget set specific, we now share all the KLoM scores for the different methods, below in Figures H.3 and H.4.

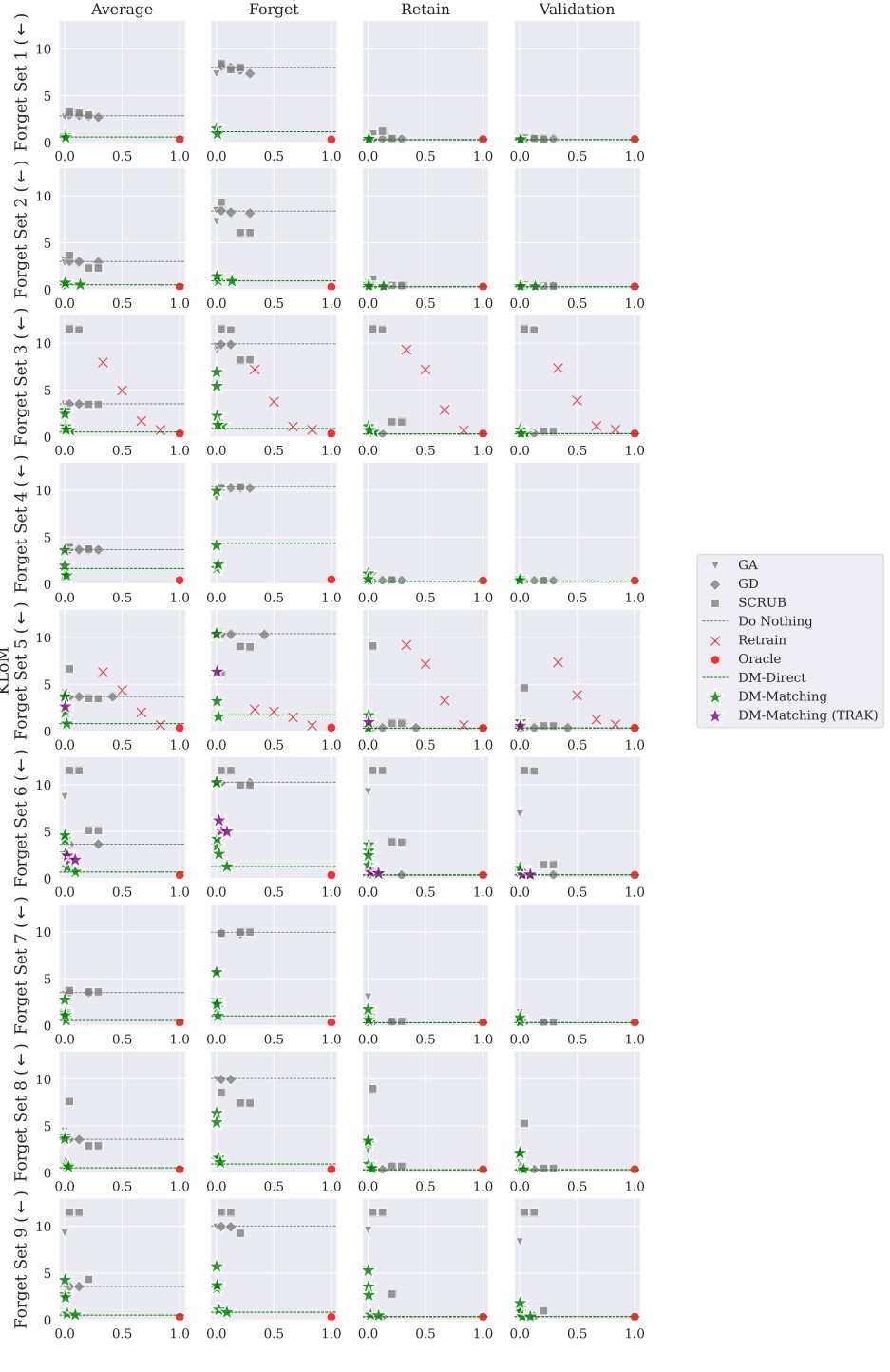

Figure H.3: KLoM results for all forget sets 1-9 on CIFAR-10. The pareto frontier for each method is in a line plot, but each KLoM data point for each method is plotted.

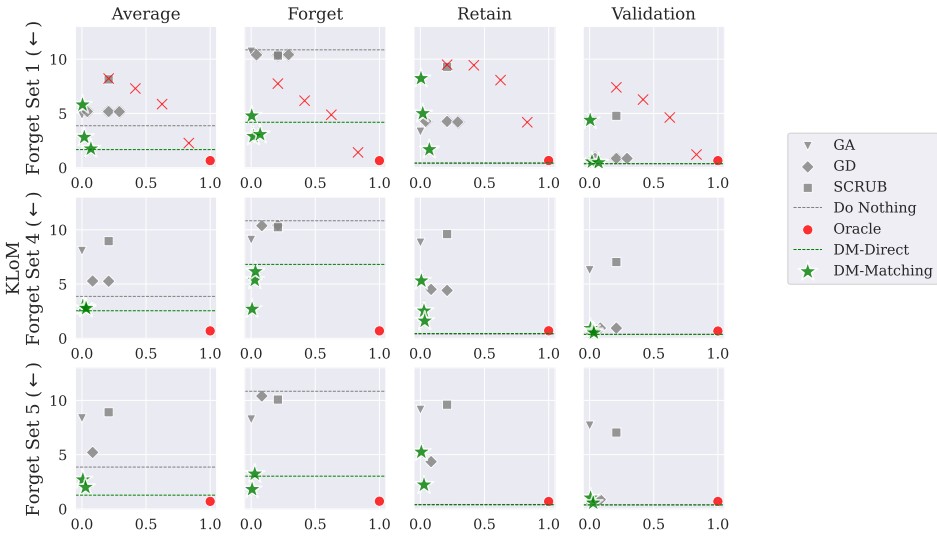

Figure H.4: `KLoM` results for all forget sets 1-3 on Living-17. The pareto frontier for each method is in a line plot, but each `KLoM` data point for each method is plotted.

Observe that Gradient Descent tends to perform very well for the retain and validation sets, but leaves the forget set relatively unchanged. This is parametrizable through hyperparameters like weight-decay, but not explored in this plot.

## H.4 MODEL ACCURACY AFTER UNLEARNING

For some measures of unlearning it is possible to do well on the machine unlearning measure-of-success, while at the same time doing completely terribly on the target task, resulting in terrible generalization and large validation error. We note that our measure of success, `KLoM`, does not suffer from this problem, as a machine unlearning algorithm is measured on its ability to match the margins of a retrained model, which is assumed to do well in general. However, for completeness, we include a table of the model accuracies for each unlearning methods, evaluated for the method that achieves the lowest `KLoM` 99th-percentile score on the forget set.

| Index | GA | GD | Scrub | Retrain an Oracle | DM-Direct | DM-Matching |
|-------|-------|-------|-------|-------------------|-----------|-------------|
| 1 | 88.34 | 88.44 | 88.29 | 88.48 | 89.97 | 89.90 |
| 2 | 88.48 | 88.49 | 88.36 | 88.51 | 89.98 | 89.70 |
| 3 | 88.25 | 88.48 | 87.55 | 88.25 | 89.96 | 89.61 |
| 4 | 87.85 | 88.44 | 88.26 | 88.53 | 89.95 | 89.46 |
| 5 | 88.07 | 88.48 | 81.99 | 88.51 | 89.93 | 89.86 |
| 6 | 84.20 | 88.47 | 85.65 | 88.42 | 89.86 | 89.47 |
| 7 | 87.98 | 88.44 | 88.26 | 88.49 | 89.98 | 89.88 |
| 8 | 88.30 | 88.49 | 81.62 | 88.43 | 90.03 | 89.93 |
| 9 | 10.92 | 88.48 | 85.36 | 88.46 | 89.99 | 89.89 |

Table H.1: Model Accuracy Results

We note that the pareto optimality curve is computed separately for each forget set. This is because because we observe that unlearning is quite model-specific and forget-specific; as such, it's reasonable that a practitioner should have a different set of hyperparameters for each unlearning depending on the forget set. Thus, having the pareto-frontier be specified per-forget-set gives each unlearning method the best opportunity to perform in that setting.

