# OpenReview forum: "Machine Unlearning via Simulated Oracle Matching"
_ICLR.cc/2025/Conference — ICLR 2025 Poster_

### Official Review · Reviewer_JhM3 · 2024-11-03

**Soundness:** 3
**Presentation:** 3
**Contribution:** 2
**Rating:** 6
**Confidence:** 5

**Summary:**

This paper introduces the concept of Oracle Matching to significantly reduce the time complexity of unlearning processes, achieving this in a fraction of the time required for traditional retraining or fine-tuning while maintaining model performance as close as possible to that of full retraining. The authors utilize the concept of "DataModels" to efficiently approximate a proxy for the oracle, leveraging this proxy within the Oracle Matching framework.

**Strengths:**

The proposed approach is both innovative and highly impactful, offering substantial reductions in unlearning time. The authors demonstrate a profound understanding of unlearning methods based on gradient descent and fine-tuning. They delve into the challenges of these methods, particularly examining their impact on the model's performance after unlearning. Although gradient-based approaches are among the most effective unlearning methods, they often degrade the predictive performance of the model considerably—a drawback that the authors have thoroughly investigated and discussed.

The concept of Oracle Matching, combined with the use of DataModels, greatly reduces time complexity. This approach shows considerable promise for improving the efficiency of unlearning processes without compromising model performance.

**Weaknesses:**

Grammatical errors:
- Abstract - line 2
- The abstract and introduction was written in rush. please


Early introduction of specific notions:
- such as $S$, $S_R$ in the beginning of the paper without providing the backgrounds and clearly stating them, confuses the reader. (Intro - second paragraph)
- line 74 - trained model $\theta$

Ambiguity:
- line 49 - Simple models // what is considered to be a simple model?
- line 59 - variety of empirical evaluations and benchmarks // what are these benchmarks? either needs to mention at least one of evaluation criteria, or rephrase the sentence.

Incorrect Statement:
- line 63 - , fine-tuning-based methods typically employ.... - incorrect, the simple fine tuning only focus on the remaining datapoints and fine tune the model on $S_R$. If there is a paper that conducts the fine tuning in the way you mentioned in this line, you need to point it out, but otherwise, check this paper ["Model Sparsity Can Simplify Machine Unlearning"](https://openreview.net/pdf?id=0jZH883i34)

Incorrect notation:
- line 152 - the notation for approximate unlearning is the same as exact unlearning.

line 83 - Empirically, we find that .... - I was expecting to see a comparison between your method and unlearning distillation approach, but you didn't made that comparison.

**Questions:**

line 246 - specify $\text{safe}(S_F)$ - how do you know this safe(S_F)? How did you calculate this?

page 6 - line 317 - this description is clear but what exactly distinguishes your approach from distillation. The comparison between your method and a distillation unlearning approach such as [Efficient Two-stage Model Retraining for Machine Unlearning](https://openaccess.thecvf.com/content/CVPR2022W/HCIS/papers/Kim_Efficient_Two-Stage_Model_Retraining_for_Machine_Unlearning_CVPRW_2022_paper.pdf)

Figure 3 - For this evaluation, did you retrain the model and then computed the unlearned model using oracle matching?

line 365: in this formulation, to create the datamodels for remaining data, you used the datamodel of whole dataset minus the datamodel of forget set. if my understanding is correct, doesn't it add the additional computation because you need to estimate two datamodels first.
also can you prove this formulation result in an exact or good estimation of datamodel for remaining data?

line 374: the true oracle output - I am just worried about the closeness of the proxy to actual retain datamodel to the.

line 404 - , the datamodel generalizes well to new forget sets in practice. very interesting how do you demonstrate this ?

ORACLE MATCHING FOR LINEAR MODELS: my understanding  is that the quality of unlearning for the oracle matching heavily is influenced by the unlearning quality of approximated oracle model . it would be interesting to investigate the influence of that model.

---

> ### Author Response · Authors · 2024-11-20
>
> We thank the reviewer for their detailed and constructive feedback, which we believe will significantly strengthen the manuscript. We point the reviewer to the general comment above, and we address each question & comment individually below.
>
> > Early introduction of specific notions:
>
> Thank you for pointing out these confusions! We have clarified the notation in our new draft. In particular, we explicitly define S in the 2nd paragraph of the paper, and make it clear even earlier that \theta are the parameters of a machine learning model.
>
> > Simple models // what is considered to be a simple model?
>
> Here we mean convex models (i.e., machine learning models for which the sample loss function is convex in the parameter vector theta). We will clarify this in our revision.
>
> > line 59 - variety of empirical evaluations and benchmarks // what are these benchmarks? either needs to mention at least one of evaluation criteria, or rephrase the sentence.
>
> We appreciate the reviewer’s comment here—we have added a sentence discussing one such evaluation (U-LiRa, which we also discuss later in the paper). We also have an extensive discussion of prior evaluation in Section 3 and Appendix D.
>
> > Grammatical errors
>
> We have double checked the abstract for grammatical correctness and were unable to find errors (in particular, we believe line 2 is correct— the “on” here relates the “effect of the forget set” and the “target prediction.” We are happy to incorporate any suggested reviewer edits for clarity or correctness.
>
> >  The simple fine tuning only focus on the remaining datapoints and fine tune the model on $S_R$. If there is a paper that conducts the fine tuning in the way you mentioned in this line…
>
> We appreciate the reviewers concern here and will strive to clarify the writing as well: in short, it is already well-known that running only GD on the retain set $S_R$ does not work in non-convex settings (unlike in strongly-convex settings; see also our response to Reviewer Bnzy’s question about GD). As a result,  most performant fine-tuning approaches employ some form of direct supervision to “erase” the forget points (we point to the results of the NeuRIPS unlearning challenge in 2023 [1] for a more elaborate discussion).
>
> In particular (and to answer the reviewer’s question directly), the prior SOTA unlearning method (called SCRUB) fine-tunes the model using a KL objective is equivalent to running gradient ascent on the forget set in addition to minimizing error on the retain set. Other approaches (see [1] for specific references) aim to achieve a similar effect using alternate strategies (e.g., re-initializing or adding noise to certain layers, etc.).
>
> We also thank the reviewer for their reference to the paper “Model Sparsity Can Simplify Machine Unlearning”---this paper takes a different (and incomparable) approach to unlearning that involves directly enforcing sparsity constraints. Since this requires modifying the model during its initial training (similar to SISA, https://arxiv.org/abs/1912.03817), it falls outside of the scope of the unlearning methods we consider.
> [1] “Are we making progress in unlearning? Findings from the first NeurIPS unlearning competition” https://arxiv.org/abs/2406.09073
>
> > Incorrect notation: the notation for approximate unlearning is the same as exact unlearning.
>
> Definition 1 and Definition 2 share notation for U (the unlearning algorithm) and for the training sets, as these refer to the same objects across definitions—in both cases, U is an algorithm mapping a machine learning model and a forget set to a new “unlearned” model. Crucially, any unlearning algorithm U that satisfies Definition 1 also satisfies Definition 2, as Definition 1 is exact equality, whereas Definition 2 is an approximate equality with epsilon-delta terms.
> From a practical perspective, the first definition limits us to approaches that structurally enforce “perfect” unlearning (e.g., as in SISA, where one by design has access to models that never saw the target forget set), which severely restricts the class of algorithms we can consider. In contrast, Definition 2 allows us to evaluate more heuristic approaches (where the model technically was trained on the forget set to start with but we attempt to unlearn it afterwards) as well as allowing us to trade-off quality vs computation time (as Fig 1 reflects).

---

> ### Author Response · Authors · 2024-11-20
>
> > line 83 - Empirically, we find that .... - I was expecting to see a comparison between your method and unlearning distillation approach, but you didn't made that comparison.
>
> We appreciate the reviewer’s comment here, and would like to take the opportunity to disambiguate our “Oracle Matching” approach from “distillation” in its most common form. In particular, rather than distilling the original model using the entire retain set, here we are distilling the oracle model using a small subset of the data (including both forget and retain points).
> We also do *not* have access to the target model in our setting (we explore this possibility in S4.2 as a proof of concept, but later remove that assumption in 4.3 and 4.4; our final DMM algorithm does not require access to a trained oracle model and is thus not direct distillation).
>
> > line 246 - specify safe(SF) - how do you know this safe(S_F)? How did you calculate this?
>
> In line 246, safe($S_F$) is just a conceptual distribution that is declared by the unlearner, and so it is completely up to the unlearning algorithm to choose its own safe($S_F$). For example, the unlearner can set safe($S_F$) to be the distribution of models trained on $S_R$ (which is what we do in our paper, and the most common approach in unlearning), but it could also do something more complex, for example, by letting safe($S_F$) be the distribution of *ensembles* of models trained on $S_R$. Conceptually, safe($S_F$) generalizes the target of unlearning, allowing the unlearner to specify a wider range of potential “unlearned” distributions. Practically, safe($S_F$) just forms the basis of the “null hypothesis” against which we will evaluate the unlearning method. The important thing is that safe($S_F$) does not depend on the forget set, and so any model from the  safe($S_F$) distribution can be said to have forgotten $S_F$.
>
> > page 6 - line 317 - this description is clear but what exactly distinguishes your approach from distillation. The comparison between your method and a distillation unlearning approach such as Efficient Two-stage Model Retraining for Machine Unlearning
>
> We thank the reviewer for this reference to the related work “Efficient Two-stage Model Retraining for Machine Unlearning” [1]. Our approaches, despite both referencing the term “distillation” are actually very different from one another and hence somewhat incomparable. We provide a brief description of the difference below, and will also disambiguate the two approaches in our related work section—please let us know if the distinction between the two methods is not clear from our description below.
> First, both the starting model and the target of distillation are completely different: In oracle matching/datamodel matching, we start with our original model (trained on the full dataset) and aim to "distill" the true retrained model. By contrast, [1] first applies an unlearning algorithm to "neutralize" the impact of the forget, and then aims to distill the original model to this neutralized model.
> Second, we finetune the original model on a small subset that *includes* the forget set by design. In contrast, the “knowledge distillation” stage [1] explicitly fine-tunes the model only on the retain set, since their target model was trained on the forget set.
>
> Finally and most importantly, note that our final algorithm (DMM) combines Oracle Matching with *datamodel-predicted* outputs, and thus does not involve any sort of distillation from another model in its final form.
> These differences make it difficult to directly compare DMM with the approach in [1].
>
> [1] “Efficient Two-stage Model Retraining for Machine Unlearning”
>
> > Figure 3 - For this evaluation, did you retrain the model and then computed the unlearned model using oracle matching?
>
> Yes this is exactly correct! To evaluate our method we compute “perfect” unlearned models by retraining from scratch on just the retain set, and evaluate the distributional difference between these perfect models and the models obtained by other methods. (Note that this retraining is only necessary for evaluating the unlearning algorithm, and is actually *more* computationally intensive than the unlearning algorithm itself, which does not require full re-training).

---

> ### Author Response · Authors · 2024-11-21
>
> > line 365: in this formulation, to create the datamodels for remaining data, you used the datamodel of whole dataset minus the datamodel of forget set. if my understanding is correct, doesn't it add the additional computation because you need to estimate two datamodels first. also can you prove this formulation result in an exact or good estimation of datamodel for remaining data?
>
> We are not 100% sure that we understood the reviewer’s point correctly, so please feel free to clarify if we have misunderstood. While it is true that implementing our approach requires estimating _several_ datamodels $\hat{f}_x$, as we note in our paper this generally involves a fixed amount of upfront computation, after which computing new datamodels is extremely straightforward/trivial. Thus, the need for multiple datamodels does not impose any extra computational burden. Furthermore, note that for a fixed target example $x$, evaluating the datamodel for two different counterfactual training sets $\hat{f}_x(S_1)$ and $\hat{f}_x(S_2)$ is extremely straightforward and involves no heavy computation (in fact, it is just a simple dot product between the estimated coefficients of $f_x$ and the corresponding set).
>
> > line 374: the true oracle output - I am just worried about the closeness of the proxy to actual retain datamodel to the.
>
> The reviewer’s comment is cut off here, but assuming the concern is about the accuracy of the datamodel oracle predictions. This is a valid concern, and indeed, the distributions in Figure 4 show that datamodel predictions do not *exactly* match the oracle outputs. However, our results from Figure 1 (and in particular, the DM-Direct algorithm) show that swapping out oracle outputs with datamodel predictions is “close enough” to be a strong unlearning algorithm—that is, the datamodel predictions are “close enough” to oracle outputs to be statistically indistinguishable.
>
> > line 404 - , the datamodel generalizes well to new forget sets in practice. very interesting how do you demonstrate this ?
>
> Here we rely on two things. First, the extensive analysis of datamodels performed by prior work, e.g., [1] and [2], which suggest that datamodels can estimate the change in model output on new counterfactual datasets (which is the key primitive for “unlearning” these different non-random subsets) very well. Second, the success of our DM-Direct algorithm across a wide range of forget sets (which just uses datamodel predictions in place of oracle outputs) indicates that datamodels are good enough predictors of model performance to be indistinguishable from true model performance in the context of unlearning.
>
> [1] Datamodels: Predicting Predictions from Training Data. https://arxiv.org/abs/2202.00622
>
> [2] TRAK: Attributing Model Behavior at Scale. https://arxiv.org/abs/2303.14186
>
> > ORACLE MATCHING FOR LINEAR MODELS: my understanding is that the quality of unlearning for the oracle matching heavily is influenced by the unlearning quality of approximated oracle model . it would be interesting to investigate the influence of that model.
>
> We thank the reviewer for this excellent suggestion. Indeed, investigating the importance of perfectly approximating the oracle model is a very interesting direction for future work. For instance, we can analyze the degree to which the convergence is influenced by the approximation error, and also study how common approximations to the oracle outputs affect the results of our analysis (we highlight more such directions in the general response above). In general our analysis only scratches the surface of interesting questions here, and we hope it motivates future work to study unlearning in this interesting setting. We leave a detailed theoretical analysis to future work, and add a discussion of this avenue to our discussion and limitations section.

---

> > ### Comment · Reviewer_JhM3 · 2024-11-26
> >
> > I thank the authors for their rebuttal and would like to maintain my original score.

---

### Official Review · Reviewer_Bnzy · 2024-11-04

**Soundness:** 3
**Presentation:** 3
**Contribution:** 3
**Rating:** 6
**Confidence:** 4

**Summary:**

This paper introduces a new Unlearning algorithm called Datamodel Matching (DMM) to unlearn a forget set by fine-tuning a model that has been pre-trained on a larger dataset including the forget and retain set. They use predictive data attribution to approximate the oracle model---the one that is retrained from scratch on the retain set and hence has not seen the forget set at all. Predictive data attribution learns datamodels for each input x to simulate how a model trained on the retain set would behave on x. With this approximation, DMM then applies Oracle Matching to align the model's output distribution with that of the oracle. They introduce KLoM for measuring the unlearning quality and show empirically that their algorithm outperforms previous gradient-based algorithms, achieving a lower KLoM and quickly approaching oracle-level accuracy with only a fraction of the retraining time.

**Strengths:**

1. The paper is well-written and easy to follow.

2. It provides a solid introduction to Unlearning, offering a detailed overview of related work, recent advancements, and existing challenges. The motivation is well-described, and the flow effectively positions this paper within the broader field, helping readers understand its scope and contributions better. The use of data attribution to approximate the Oracle model is particularly compelling.

**Weaknesses:**

1. The experimental results, while supportive of the algorithm’s effectiveness, are somewhat limited in scope. The analysis focuses primarily on CIFAR-10 and an ImageNet subset (Living-17), with figures 1 and 3 illustrating outcomes only on these datasets. Expanding the experiments to include additional datasets would enhance the generalizability of the findings. On the Retain set, Oracle Matching gets close to the Oracle in CIFAR-10 but this is not the case with Living-17 which means that while the algorithm can reduce KLoM on the forget set, it does not work as well on the retain set. More discussion on this observation can improve the understanding of the algorithm’s limitations, and I'm interested to know how this observation extends to other tasks.

2. The paper could benefit from clearer explanations and interpretations of the figures and baselines. The discussion around interpreting each figure is limited. Additionally, some of the baseline methods are not well-defined; for instance, SCRUB is introduced without a description, and GD (Gradient Descent) is not explained well until page 9 (also see my question on GD in the Questions).

3. Minor typo: The RHS of equation (2) should be dependent on x.

4. Calculating one $\beta$ vector for each x in the dataset appears computationally intensive. Although the authors mention that this is a one-time process that amortizes over unlearning requests, further discussion on its computational cost relative to the unlearning phase would be helpful.

**Questions:**

1. This question is regarding the GD baseline. Given a model $\theta_{full}=A(S)$ trained on a full dataset S (including the retain set $S_R$), GD minimizes the loss on the retain set using gradient descent, starting from $\theta_{full}$. Since $S_R$ was already in the dataset S that the model was trained on, this essentially involves further training on $S_R$, which could lead to overfitting without effectively forgetting $S_F$. This may explain why, in Figure 3, GD performs similarly to 'Do Nothing' and, on average, even worse. My interpretation is that further training on $S_R$ may not significantly alter a model that was sufficiently trained on S if the loss is strongly convex and could lead to overfitting (and hence degrading performance on the validation) in more complex landscapes. Could the authors clarify if there’s any specific benefit of GD for unlearning that I might be overlooking?

2. The authors mention the possibility of having duplicates across the forget and retain sets on page 6 when discussing the drawbacks of Gradient Ascent. Given that $S_F$ and $S_R$ are sets and $S_R$ is defined as $S \backslash S_F$, I don't see how this duplication is possible. Could the authors clarify this?

3. Could the authors provide more interpretation of the results in Figure 2? How do you interpret the changes and fluctuations in the red and gray lines?

---

> ### Author Response · Authors · 2024-11-21
>
> We thank the reviewer for their detailed and constructive feedback, it is highly appreciated and will improve the manuscript. We point the reviewer to the general comment above, and address their questions individually below.
>
> > The experimental results, while supportive of the algorithm’s effectiveness, are somewhat limited in scope. The analysis focuses primarily on CIFAR-10 and an ImageNet subset (Living-17), with figures 1 and 3 illustrating outcomes only on these datasets. Expanding the experiments to include additional datasets would enhance the generalizability of the findings.
>
> We agree that more evaluations would be valuable. As our main goal is to present the general conceptual framework and accompanying theoretical analysis, we focused our evaluation on two datasets. We are currently running our method on a new dataset (text classification) and will include the results in our camera ready version, but due to compute constraints it is very unlikely that the results will be ready by the end of the discussion period (see also the general comment).
>
> Note that even in the two settings where we did evaluate our algorithm, our evaluations are still *significantly* stronger than prior work in similar settings, as we evaluate on more challenging (small & non-random) forget sets and also use pointwise evaluations (which as prior work [0] found is important to detect failure modes).
>
> Finally, we have good reason to believe that the algorithm will continue to succeed in settings where good data attribution methods are available. In particular, the success of our framework depends on two factors: (i) fidelity of the datamodel approximation and (ii) how successfully OM can navigate the optimization landscape. Re (i): although we only evaluate on these two datasets, prior work shows that these algorithms perform well on larger-scale settings and across various modalities (text classification and CLIP [1], language modeling [2], diffusion [3,4]). Re (ii), our theoretical analysis in Sec 6 further corroborates the empirical success of OM. Together, these findings suggest that DMM will continue to  succeed in other settings (different modalities, etc.).
>
> [0] Inexact Unlearning Needs More Careful Evaluations to Avoid a False Sense of Privacy. https://arxiv.org/abs/2403.01218
>
> [1] TRAK: Attributing Model Behavior at Scale. https://arxiv.org/abs/2303.14186
>
> [2] Training Data Attribution via Approximate Unrolled Differentiation. https://arxiv.org/abs/2405.12186
>
> [3] The Journey, Not the Destination: How Data Guides Diffusion Models. https://arxiv.org/abs/2312.06205
>
> [4] Influence Functions for Scalable Data Attribution in Diffusion Models. https://arxiv.org/abs/2410.13850
>
> > On the Retain set, Oracle Matching gets close to the Oracle in CIFAR-10 but this is not the case with Living-17 which means that while the algorithm can reduce KLoM on the forget set, it does not work as well on the retain set. More discussion on this observation can improve the understanding of the algorithm’s limitations, and I'm interested to know how this observation extends to other tasks.
>
> We thank the reviewer for this comment, and we expand on the response below in our new discussion and limitations section.
>
> Indeed, there is a larger train-test gap (i.e., overfitting) on Living-17 than CIFAR-10 for the particular hyperparameter settings we used. We suspect that this is why OM (and all gradient-based methods we tried) does worse on the retain set; fine-tuning with OM inadvertently “reverses” some of the overfitting. Exploring ways to regularize OM to prevent this (e.g., using heavy l2-regularization towards the original model in parameter space) is an interesting direction for future work.
>
> That said, note that this gap between retain and validation set (which, recall, is simply a train-test/overfitting gap) is potentially less of an issue in more practically relevant “modern” training regimes; for example, modern LMs are typically trained for ~one epoch and show no overfitting (train = test loss). We thus suspect that the retain set issue would vanish in these settings: further evaluating our approach in such settings to confirm or rebut this intuition would be another valuable avenue for future work.
> (continued below)

---

> ### Author Response · Authors · 2024-11-21
>
> As an initial exploration (see updated Appendix G.1), we investigated why there was significant overfitting on Living17, and found the lack of use of data augmentation to be the main cause (the higher-dimensionality of images compared to CIFAR-10 also likely exacerbating this). We had turned off data augmentation for all of our experiments as this was done on all major prior works on (empirical) unlearning for DNNs. Under new hyperparameter settings that are identical except the use of standard image data augmentation (random cropping and flips), the trained models overfit much less. In this new setting, we re-trained oracle models and applied oracle matching with the same exact hyperparams as before (so they are not optimized for the new setup). The resulting KLoM scores on the retain set are significantly lower (95% decreasing from 2.88 to 0.62, supporting our hypothesis that significant overfitting caused OM to perform worse on the retain set on Living17.
>
> > The paper could benefit from clearer explanations and interpretations of the figures and baselines...
>
> Thank you for the feedback. We added more details to the figure captions and also added a description of each baseline to Appendix C and point to it from Section 4.
>
>
> > Minor typo: The RHS of equation (2) should be dependent on x.
>
> We believe that the reviewer means to point out the typo on the LHS, which we have now edited to be $f^hat_x(S’)$, rather than $f^hat(S’)$ (thank you!). However, if you believe there is an issue on the RHS as well, please let us know - we thank the reviewer for their careful reading.
>
> > Calculating one β vector for each x in the dataset appears computationally intensive. Although the authors mention that this is a one-time process that amortizes over unlearning requests, further discussion on its computational cost relative to the unlearning phase would be helpful.
>
> The costs of estimating datamodels (with the methods used in this paper) are as follows:
> - For the regression-based approach (S4.3-4.4): ~1000s of re-trainings
> - For TRAK (S5.2; and similar algorithms in the literature), O(1) re-trainings
>
> Hence, amortized over many unlearning requests, using DMM with the second approach would be much (significantly) cheaper than re-training. The first approach arguably remains impractical, but demonstrates that we can achieve close to optimal unlearning using a sufficiently good oracle. This implies that any improvements to the predictiveness or efficiency of data attribution methods would directly translate to more accurate and faster unlearning with DMM, helping make it more practical.
>
> > This question is regarding the GD baseline. Given a model θ_full=A(S) trained on a full dataset S (including the retain set S_R), GD minimizes the loss on the retain set using gradient descent, starting from θ_full. Since S_R was already in the dataset S that the model was trained on, this essentially involves further training on S_R, which could lead to overfitting without effectively forgetting S_F....Could the authors clarify if there’s any specific benefit of GD for unlearning that I might be overlooking?
>
> The intuition provided by the reviewer is exactly correct in every case *except* the strongly convex setting---and it is also precisely why GD is a bad baseline for the deep learning setting, despite its popularity. In particular (as the reviewer points out), in overparameterized regimes, it’s hard to force the model to “forget” examples it has already fit without explicit guidance (as done by OM or GA on the forget set).
>
> We include GD as a baseline, because it is a popular baseline in prior work on unlearning, even in the non-convex setting. The reason why GD is used as a baseline at all is likely due to its success in the *strongly convex* setting, where the intuition given by the reviewer actually does not hold: since strongly convex loss functions have a unique minimizer, simply running GD on the retain set alone will provably lead to reaching the oracle model (the proof of this is quite simple—GD on a strongly convex loss function must reach the unique minimizer, regardless of where we initialize, and so eventually in the strongly convex setting GD must “forget” the forget set points). For analysis on GD as an unlearning algorithm in the convex case we refer the reader to [1].
>
> To summarize our answer: GD *is* actually a valid (and strong) unlearning algorithm in the strongly convex case. However, since deep learning is not strongly convex, the guarantees around GD break down which likely cause it to perform similarly to (or worse than) “Do Nothing.”
>
> [1] "Descent-to-Delete: Gradient-Based Methods for Machine Unlearning" (https://arxiv.org/abs/2007.02923)

---

> > ### Author Response · Authors · 2024-11-21
> >
> > > The authors mention the possibility of having duplicates across the forget and retain sets on page 6 when discussing the drawbacks of Gradient Ascent. Given that $S_F$ and $S_R$ are sets and $S_R$ is defined as $S∖S_F$. I don't see how this duplication is possible. Could the authors clarify this?
> >
> > By duplication here, we just mean having multiple copies of the same example in the dataset $S$ (e.g., vision and language datasets tend to have many duplicate or near-duplicate samples), rather than inclusion of the exact same sample in both $S_F$ and $S_R$ (which, as the review points out, are indeed a strict partition of $S$).
> >
> > For example, if the forget set consists of images of Roger Federer playing a specific tennis match, and the retain set includes more pictures of Roger Federer from the same match, we would expect that that re-trained oracle model would still retain some semantic knowledge of that match, just based on his images in the retain set. That is, the re-trained oracle model would actually do extremely well on the images in the forget set, despite having never seen them before (just because of the semantic duplicates in the retain set). However, by explicitly running GA on the forget set, one ‘forces” the model to perform poorly on these forget set images, which is the opposite of how the oracle model would perform.
> >
> > > Could the authors provide more interpretation of the results in Figure 2? How do you interpret the changes and fluctuations in the red and gray lines?
> >
> > Fluctuations in this experiment on unlearning stability of SCRUB can be attributed to (at least) three sources. First, there is randomness introduced by stochastic optimization (we use mini-batch SGD) to optimize the SCRUB objective. Second, the SCRUB objective uses gradient ascent (GA), which as we discuss suffers from the missing targets problem, and so eventually the algorithm overshoots and the unlearning quality degrades. Finally, there are interactions between data points: as the algorithm updates the model to unlearn one point, that also influences its outputs on other points, thus possibly requiring further modifications on those other points.  We add a discussion of these fluctuations to the caption of Figure 2.

---

> > > ### Author Response · Authors · 2024-11-25
> > > **End of Review Period Approaching**
> > >
> > > Dear Reviewer, as we get to the end of the response period, please let us know if we have addressed your main concerns or if there are issues you'd like to discuss further, and if not, please consider raising your score. Thank you in advance!

---

> ### Comment · Reviewer_Bnzy · 2024-12-03
>
> I thank the authors for addressing my questions and for adding a discussion and limitations section to the paper, which I believe improves the manuscript. Due to concerns regarding the scalability of the algorithm, as also mentioned in the limitations section, I refrain from increasing my score to 8. Nonetheless, I support accepting the paper as indicated by my initial score.

---

### Official Review · Reviewer_Y44J · 2024-11-04

**Soundness:** 4
**Presentation:** 4
**Contribution:** 4
**Rating:** 8
**Confidence:** 4

**Summary:**

The paper addresses the problem of _machine unlearning_ (removing the effect of a few training data points "forget set" on the model's outputs) with by reducing it to the problem of _data attribution_ (predicting the effect the training set on the model's outputs). With this, the paper proposes a meta-algorithm Datamodel Matching (DMM) that gets predictions from data attribution on all-but-forget set and finetunes the model to match the predictions. A new unlearning metric KL Divergence on Metrics (KLoM) is also introduced. Finally, the paper presents experiments on unlearning in image classification tasks, showing that DMM is better at unlearning and faster than naive-retraining on the all-but-forget-set.

**Strengths:**

The paper is written extremely well, and has a natural flow to it. When reading I thought of many questions and added annotations, only to find them answered in the next paragraph or section. E.g. the authors introduce Oracle Matching with the strong assumption of oracle access to data attributor $f^{oracle}$ (Section 4.2), and immediately discuss how to simulate such an oracle without such an access (Section 4.3). In Section 5 as well, the paper attends to a natural question readers could have: is oracle matching useful when the problem is easy to solve with gradient descent. This is good writing, and I appreciate the authors' efforts in putting themselves in the readers' shoes.

In sum, the Oracle Matching and Simulation methods are intuitive yet thoroughly tested on image classification tasks. These methods are original as well, and it is surprising to me that linear datamodels work so well empirically. I'd like to see this paper accepted, and the research directions it inspires.

**Weaknesses:**

It is unclear that linear datamodels extend to other kinds of tasks, e.g. language modeling or regression problems. I believe this to be a major weakness of the paper. While linear datamodels lead to simple algorithms in this paper, the previous work [1] does not have a good argument for why linear datamodels work [1; Section 7.2]---in fact Figure 6 of [1] display imperfect matching using linear datamodels. It'd be useful to mention this limitation in this manuscript as well, and discuss the limitation's impact to machine learning.

# Suggestions:
1. Line 156. It'd be useful to the reader to add a citation on differential privacy, e.g. one of the standard works like [2].
2. Line 176. $\hat{f}$ should have output range in $\mathbb{R}^k$ since the range of $f_x$ is in $\mathbb{R}^k$.
3. Line 182. "show" -> "empirically show".
4. Definition 3. Write safe, $S_F$, and input $x$ explicitly in KLoM, otherwise KLoM$(\mathcal{U})$ looks like KLoM of the unlearning function across _all_ safe functions and inputs. I'm curious why the authors wrote KLoM$(\mathcal{U})$.
5. Add a Limitations section.

[1] Ilyas, A., Park, S. M., Engstrom, L., Leclerc, G., & Madry, A. (2022). Datamodels: Predicting predictions from training data. arXiv preprint arXiv:2202.00622.
[2] Dwork, C., & Roth, A. (2014). The algorithmic foundations of differential privacy. Foundations and Trends® in Theoretical Computer Science, 9(3–4), 211-407.

**Questions:**

# High-level questions
1. Definition 3. Why is KLoM called KL-divergence of Margins? I couldn't find the reasoning.
2. Line 321 and Alg A.1. Assuming that dataset $S$ has distinct datapoints, isn't $S_{finetune}$ the same as $S$, implying that Alg A.1 is simply matching the model's outputs to the oracle? Then I don't understand the novelty of Oracle Matching.
3. Line 364. Why replace the first term with $\beta$? Linear $\beta$ is exactly the definition of the estimator $\hat{f}$, which is only an approximation to $f_x$. It'd make sense to explicitly state that DM-DIRECT approximately simulates the oracle outputs. You could then argue that even this approximate simulator works well empirically.

# Low-level questions
1. Line 173. Is this estimator/datamodel only for a single $x$? Does this mean that distinct inputs might require distinct datamodels?
2. Equation 2. What is the approximation in? Is it in some measure of distributions?
3. Figures. Why are there multiple points on the plot for each legend entry?

---

> ### Author Response · Authors · 2024-11-20
>
> We thank the reviewer for the detailed and constructive feedback, as well as the valuable questions and suggestions for improvement. We address each point individually below, and also refer the reviewer to our general response above.
>
>
> > It is unclear that linear datamodels extend to other kinds of tasks….
>
> We appreciate the reviewer’s point that linear datamodels will not always be a perfect predictor of model behavior, and discuss this fact in our new limitations section (see the general comment). Even so, we believe our method and framework is useful for two reasons:
>
> First, our primary contribution in this work is a general “reduction” from the problem of unlearning to the problem of data attribution. Importantly, this reduction does not depend on linearity of the datamodel at all—as long as one has an easy-to-evaluate datamodel that maps from training subset to (predicted) model behavior, one can plug in that datamodel to yield a strong unlearning algorithm. Thus, as researchers develop better (more predictive or more efficient) or even non-linear attribution methods, we can directly plug them into our meta-algorithm to get better unlearning algorithms. Overall, we find it encouraging that even using this “crude” linear approximation for DMM still leads to SOTA unlearning performance that vastly outperforms all prior methods.
>
> Second, even linear datamodels are surprisingly effective for various modalities. While our evaluations are limited to image classification models, prior works show that  linear datamodels are also effective for text classification [1], language modeling [1,2], and even diffusion models [3,4]. While the origin of this linearity is not fully understood yet, [5] provides some intuition in a theoretical model to suggest why linear models may be surprisingly effective.
>
> We again thank the reviewer for highlighting this important point and we discuss this limitation more extensively in the revision.
>
> [1] TRAK: Attributing Model Behavior at Scale. https://arxiv.org/abs/2303.14186
>
> [2] Training Data Attribution via Approximate Unrolled Differentiation. https://arxiv.org/abs/2405.12186
>
> [3] The Journey, Not the Destination: How Data Guides Diffusion Models. https://arxiv.org/abs/2312.06205
>
> [4] Influence Functions for Scalable Data Attribution in Diffusion Models. https://arxiv.org/abs/2410.13850
>
> [5] Saunshi et al. Understanding Influence Functions and Datamodels via Harmonic Analysis. https://arxiv.org/abs/2210.01072
>
> > Line 156. It'd be useful to the reader to add a citation on differential privacy, e.g. one of the standard works like [2] (Dwork paper).
> > Line 176. \hat{f}  should have output range in R^k since the range of  f_x is in Rk
> > Line 182. "show" -> "empirically show".
> > Definition 3. Write safe(SF) and input x explicitly in KLoM, otherwise KLoM(U) looks like KLoM of the unlearning function across all safe functions and inputs. I'm curious why the authors wrote KLoM(U).
>
> > Add a Limitations section.
>
> We thank the reviewer for the valuable suggestions and thorough reading of our paper—we have fixed all of the suggested line edits. As discussed in the general commend, we will also add a limitations section that discusses (a) potential misspecification in datamodels, (b) computational challenges, and (c) challenges to scaling with the number of target classes.
>
>
> > Definition 3. Why is KLoM called KL-divergence of Margins? I couldn't find the reasoning.
>
> We call our metric KL-divergence-of-Margins because it (a) first computes the distribution of classification margins (“margins” as defined in https://arxiv.org/abs/2403.01218 and https://arxiv.org/abs/2112.03570) for both unlearned models and oracle models; then (b) measures the KL divergence between these two distributions. Overall, we are measuring the KL divergence between two distributions of margins (unlearned vs the oracle ground-truth), and hence the name KLoM.

---

> ### Author Response · Authors · 2024-11-20
>
> > Line 321 and Alg A.1. Assuming that dataset S has distinct datapoints, isn't S_finetune the same as S, implying that Alg A.1 is simply matching the model's outputs to the oracle? Then I don't understand the novelty of Oracle Matching.
>
> We construct the finetuning set $S_{finetune}$ by combining the forget set with a *subsample* of the retain set; hence, $S_{finetune}$ is not the same as the full train set $S$. The point is that we can finetune on a small fraction of $S$ and still effectively “distill” the oracle model. That said, even in the case that $S_{finetune} = S$, it is not obvious that OM will converge *quickly*, which is a key finding of our paper (indeed, convergence on its own is not surprising or novel). For example, if $\theta$ is far (in parameter space) from an “oracle” model, then SGD with small learning rate will not be able to converge to an oracle model quickly. Our empirical results in Section 4.2 and 4.4 and our theoretical analysis in Section 5, however, show that starting from a fully trained model, one can converge *quickly* to an oracle model by running OM. Furthermore, even in the case $S_{finetune}=S$, it is not obvious that our unlearned model will generalize *out of sample*, which is important for matching the behavior of the retrained model on validation points.
> Conceptually, the success of OM demonstrates that the underlying issue with existing fine-tuning approaches is the “missing targets” problem, rather than a fundamental difficulty with optimization.
>
> > Line 364. Why replace the first term with β?...
>
> Indeed, the reviewer has caught onto a subtle point that we will also elaborate on more thoroughly in our revision of the manuscript. The reason for replacing the first term with $f_x(\theta_0)$ is three-fold:
> - First (and most basically), it removes the need to estimate an implicit bias term or intercept for the linear datamodel.
> - Second (and slightly more subtly), it also improves the performance of our estimator—since the forget set is typically much smaller than the retain set, and our estimates of the coefficients $\beta$ are likely to be imperfect, the variance of our revised estimator will be significantly smaller than the sum over the retain set (where errors in the coefficients may compound).
> - Finally, one can view the “original” formulation (i.e., the sum over the retain set) as predicting the output of the *average* model when training on the retain set $S_R$ (where the average is taken over all other sources of randomness). Our goal, however, is to predict the output of the specific base model had it not been trained on $S_F$—our “workaround” is thus to estimate the average *delta* in output incurred from not training on $S_F$, and subtract that off of the current output of the base model in order to estimate this counterfactual value.
>
> > Line 173. Is this estimator/datamodel only for a single x? Does this mean that distinct inputs might require distinct datamodels?
>
> That is correct, and is consistent with prior work on data attribution. This design is by choice, as different inputs depend on the training data in a different way and therefore require different datamodels. Note that typically most of the estimation cost for datamodels comes from a “bulk” computation independent of target example, so this can be amortized and the additional cost for a new x is marginal.
> - For example, in the regression-based estimator, the bulk of the computation involves training models on different subsets of the training set and evaluating them; subsequently estimating a datamodel for a new input simply involves fitting a linear regression.
> - As another example, in the TRAK estimator and similar variants, the bulk of the computation involves computing gradients on the full dataset; subsequently estimating a datamodel involves a few matrix operations on those gradients and hence is cheap.
>
> > Equation 2. What is the approximation in? Is it in some measure of distributions?
>
> The definition is somewhat general on purpose, as we make things more concrete later in the paper—one could for example take $\hat{f}$ to be the choice that minimizes the MSE between $\hat{f}$ and $f$ over some distribution over subsets $S’ \subset S$ (as was done by [2]), but in practice all we want is for $\hat{f}$ to be a sufficiently accurate proxy (in prediction space) to the true underlying function f to enable approximate unlearning (as defined in Section 2).

---

> > ### Author Response · Authors · 2024-11-20
> >
> > > Figures. Why are there multiple points on the plot for each legend entry?
> >
> > Each point (for the same legend entry) corresponds to running the same algorithm with different compute budgets (i.e., for different amounts of time). Our goal is to understand the performance of the entire family of algorithms (e.g., GD for t steps for varying t) and understand trade-offs between compute time and unlearning quality. As Fig 2 shows, this time budget is an important hyperparameter that affects the degree of success of unlearning, but also the feasibility of the resulting algorithm. For example, simply retraining on the retain set from scratch (the point on the far right) is a “perfect” unlearning algorithm in that it achieves KLoM score 0, but does so at immense computational cost (the x axis), while re-training for limited amount of time (the other red X’s in the plot) does not yield an effective unlearning algorithm. We clarify this point in the revised captions. These types of plots are standard in work on unlearning, where the primary goal is to improve the  pareto frontier of compute vs. unlearning quality.
> >
> > Note that for baselines, we optimize the hyperparameter for each choice of t (for each t or individual point), while for our methods (OM and DMM), we only optimize the hyperparameter globally (across all t), then just simply finetune for shorter or longer duration (only varying t). This makes the evaluation strictly harder for our algorithm.

---

> > > ### Comment · Reviewer_Y44J · 2024-11-25
> > > **Reply to Rebuttal**
> > >
> > > Thank you for the clarifications. Please do add the limitations you mean to the main paper, as they will help future work. My score remains positive.
> > >
> > > - I was confused by lines 4 and 5 of Alg A.1, which say $S_R' \gets S \setminus S_F$ and $S_{finetune} = S_F \cup S_R'$. The comments mention that $S_R'$ is a subsample of r points from the retain set. But the notation is misleading here, implying that $S_{finetune} = S$ since `\setminus` ($\setminus$ operator) doesn't include the subsampling step. Could you please fix this? Also in Alg A.3. These are your main algorithms for the paper.
> > > - Out of curiosity, how does a low-variance estimator improve its performance?

---

> ### Author Response · Authors · 2024-12-02
>
> We have now added a limitations section (Appendix A) in our revision---thank you for the initial suggestion.
>
> - We missed this for our rebuttal revision, but will incorporate additional notation (e.g., S_R' <- SampleRandom(S \ S_F, k))
> in our final revision to clarify this.
> - We are not sure we interpreted the question correctly, but if the question is: *does a more accurate datamodel lead to better DMM performance?* In that case, our (new) analysis in Appendix G.2 shows that as the datamodels improve in predictiveness, then the resulting DMM also performs better unlearning. If this wasn't the question, please elaborate and we are happy to discuss further.
>
> Thank you again for the constructive and encouraging feedback throughout.

---

### Author Response · Authors · 2024-11-21

We thank the reviewers for their detailed and thoughtful reviews of our work. We address each reviewer below individually, and wanted to highlight a few changes we are making to the manuscript:

- First, we are adding a discussion & limitations section that will discuss a few limitations of our algorithm, as well as interesting directions for future work. Our discussion focuses on three points brought up by the reviewers. First, the suboptimality of linear datamodels—developing accurate (potentially non-linear) attribution methods that are efficient to estimate is a key area of future work in this space. We also highlight a limitation of our approach in that naively, its complexity grows with the number of classes in the classification problem, and thus adapting it to, e.g., the language modeling setting is an important direction for future work. Finally, on the theoretical front there are a variety of questions that our analysis does not answer, such as the robustness of oracle matching to misspecification or approximation error, as well as how to optimally construct the fine-tuning set.

- Second, we are in the process of running our procedure on more datasets and models, in order to further prove the robustness of our method. While it is unlikely that these experiments will finish before the rebuttal period, we commit to including the results (whether positive or negative) in the camera-ready version of our manuscript.

We believe that both of these changes will significantly strengthen the final version of our manuscript, and again thank the reviewers for their details and thorough comments and questions. In the meantime, we will soon upload a revised draft incorporating the reviewer’s suggestions.

---

> ### Author Response · Authors · 2024-11-28
> **Revised draft**
>
> We have uploaded a revised draft incorporating the reviewer's various suggestions.
>
> Beyond local edits, we add the following new sections and experimental results:
> - Our new Appendix A discusses limitations and directions for future work
> - We included new analysis of overfitting on Living-17 in Appendix G.1, showing that under a different choice of hyperparameters with less overfitting, DMM matches oracle outputs much better on the retain set.
> - We included new analysis of how DMM performance scales with the error in datamodel estimates in Appendix G.2
>
> We hope that this revised draft aids in the remaining discussion.

---

### Meta-Review · Area_Chair_pRhD · 2024-12-20

**Metareview:**

The paper was praised for its clear presentation. The introduction of Oracle Matching and the use of data attribution to approximate the oracle model were seen as innovative and promising contributions to the field of machine unlearning. The proposed method, DMM (Datamodel Matching), demonstrates strong empirical performance in unlearning, outperforming existing gradient-based algorithms and achieving results close to the oracle model with less computational cost. The paper provides an analysis of its convergence properties and the impact of datamodel accuracy on unlearning performance.

**Additional Comments On Reviewer Discussion:**

The generalizability of linear datamodels to tasks beyond image classification was questioned, particularly for language modeling and regression.  The overfitting issue observed on the Living-17 dataset sparked a discussion about the method's limitations and potential issues when handling specific data types or model architectures.

The need for clearer explanations of figures and baseline methods was addressed by the authors.  The computational cost of the method, particularly the cost of estimating datamodels for each input, was discussed, with the authors clarifying the differences in cost depending on the specific data attribution method used.

The rebuttal period successfully addressed the reviewers' concerns and questions. Some concerns regarding scalability remained, but the overall response to the paper was positive.

---

### Decision · Program_Chairs · 2025-01-22

Accept (Poster)